RESEARCH COMMUNICATION

 

# An afferent white matter pathway from the pulvinar to the amygdala facilitates fear recognition

Jessica McFadyen[1,2]*, Jason B Mattingley[1,2,3,4], Marta I Garrido[1,2,5,6]

[1]Queensland Brain Institute, University of Queensland, Brisbane, Australia; [2]Australian Research Council of Excellence for Integrative Brain Function, Clayton, Australia; [3]School of Psychology, University of Queensland, Brisbane, Australia; [4]Canadian Institute for Advanced Research (CIFAR), Toronto, Canada; [5]School of Mathematics and Physics, University of Queensland, Brisbane, Australia; [6]Centre for Advanced Imaging, University of Queensland, Brisbane, Australia

**Abstract** Our ability to rapidly detect threats is thought to be subserved by a subcortical pathway that quickly conveys visual information to the amygdala. This neural shortcut has been demonstrated in animals but has rarely been shown in the human brain. Importantly, it remains unclear whether such a pathway might influence neural activity and behavior. We conducted a multimodal neuroimaging study of 622 participants from the Human Connectome Project. We applied probabilistic tractography to diffusion-weighted images, reconstructing a subcortical pathway to the amygdala from the superior colliculus via the pulvinar. We then computationally modeled the flow of haemodynamic activity during a face-viewing task and found evidence for a functionally afferent pulvinar-amygdala pathway. Critically, individuals with greater fibre density in this pathway also had stronger dynamic coupling and enhanced fearful face recognition. Our findings provide converging evidence for the recruitment of an afferent subcortical pulvinar connection to the amygdala that facilitates fear recognition.

**Editorial note:** This article has been through an editorial process in which the authors decide how to respond to the issues raised during peer review. The Reviewing Editor's assessment is that minor issues remain unresolved (see decision letter).

DOI: https://doi.org/10.7554/eLife.40766.001

*For correspondence:
j.mcfadyen@uq.edu.au

**Competing interests:** The authors declare that no competing interests exist.

## Introduction

Decades ago, rodent research uncovered a subcortical pathway to the amygdala that rapidly transmits auditory signals of threat even when the auditory cortex is destroyed (*Ledoux, 1998*). Since this discovery, researchers have sought an equivalent visual pathway that might explain how it is that people with a lesioned primary visual cortex can still respond to affective visual stimuli that they cannot consciously see (*Tamietto et al., 2010*). The superior colliculus, pulvinar, and amygdala have been identified as nodes of a human subcortical route to the amygdala that bypasses the cortex (*Morris et al., 1999*). These subcortical areas consistently coactivate in cortically blind patients (*Pegna et al., 2005*) – as well as in healthy adults (*Vuilleumier et al., 2003*; *Morris et al., 1999*) – when they view emotional stimuli, such as angry or fearful faces. Magnetoencephalography studies using computational modelling have investigated whether the activation of these subcortical nodes is causally related. These studies have consistently found evidence for a forward connection between the pulvinar and amygdala (*McFadyen et al., 2017*; *Garvert et al., 2014*; *Rudrauf et al., 2008*). The dynamic causal relationship between the superior colliculus and the pulvinar, however, remains unexplored in the human brain (*Soares et al., 2017*). The pulvinar also has several functional and

**eLife digest :** Being able to quickly detect and respond to potential threats is essential for survival. Fear and threat trigger a range of responses in the body, which are controlled by different regions in the brain. For example, a structure located deep within the brain called the amygdala is connected to other parts of the brain that regulate hormones, senses and muscles. The amygdala is highly responsive to signs of threat, and research in rodents has shown that it plays a role in transmitting sounds that indicate danger. However, so far it has remained unclear if this was also the case for visual information.

This is particularly challenging to study in humans because it has been difficult to image the deeper regions in the human brain. Now, McFadyen et al. reconstructed the pathways between the deeper brain regions important for processing vision and the amygdala using the brain scans of 622 participants. Then, they tested whether there was any connection between these pathways and the ability to recognise emotional expressions. To do so, fMRI brain scanning was used to measure the blood flow in the brain of volunteers looking at 40 faces that were either happy, sad, angry, fearful or neutral.

The results showed that when people were looking at pictures of fearful and angry faces, the blood flow between visual areas and the amygdala increased, especially in individuals with stronger connections, such denser nerve fibres, between the involved regions. The denser those fibres were, the better the people were at recognising when a face was fearful.

These discoveries suggest that the amygdala also plays a role in transmitting signals from deep-brain visual areas indicating danger and is likely to be one of the first areas to trigger a fear response in the brain. People with autism respond less to fearful faces, while people with anxiety respond more. Future research could investigate if the pathways to the amygdala differ in these people.

DOI: https://doi.org/10.7554/eLife.40766.002

cytoarchitectural subregions (*Barron et al., 2015*) and it is unclear how these connect to the superior colliculus and the amygdala and what roles these subregions may play in mediating transmission along the subcortical route (*Koller et al., 2018*; *Pessoa and Adolphs, 2010*). As such, the hypothesis that the subcortical route rapidly transfers information from the retina to the amygdala without interference has been heavily criticised (*Pessoa and Adolphs, 2010*; *Pessoa and Adolphs, 2011*). Furthermore, the pulvinar is highly connected with a widespread network of cortical regions that may contribute to transmission along the subcortical route (*Bridge et al., 2016*; *Zhou et al., 2016*). Hence, it remains unknown whether the functional activation of the human superior colliculus, pulvinar and amygdala during affective processing bears any relation to an underlying structural pathway (*Pessoa and Adolphs, 2010*).

Recent animal research has revealed several potential direct subcortical pathways that have a causal relationship with fearful behaviour in response to visual threats (*Zhou et al., 2017*; *Wei et al., 2015*; *Shang et al., 2015*). In the absence of relevant postmortem human research, however, our anatomical knowledge of the human subcortical route to the amygdala can only be derived from tractography of diffusion-weighted images (DWI). *Tamietto et al., 2012* examined DWIs from a blindsight patient whose left primary visual cortex was destroyed. The white matter structure of the subcortical route was estimated for the patient and for ten healthy, age-matched controls. Critically, the subcortical route had greater fractional anisotropy in the patient's damaged hemisphere, suggesting a neuroplastic increase in structural connectivity to compensate for the disrupted cortical pathways (*Tamietto et al., 2012*). In a similar study, *Rafal et al. (2015)* used tractography to investigate the subcortical route in 20 healthy humans and eight macaques. The subcortical route was reconstructed in both hemispheres for 19 of the 20 human participants and 7 of the eight macaques (*Rafal et al., 2015*). Notably, this sample of human participants was recently expanded and re-examined, further demonstrating that individuals with greater fractional anisotropy along the subcortical route also had a stronger bias toward threat when making saccades to scenes (*Koller et al., 2018*).

Diffusion tractography may grant insight into the strength of anatomical connectivity between regions, but it cannot reveal the direction of information transfer nor can it be used as direct

evidence alone for the anatomical existence of a neural pathway. The anatomical presence and the direction-specific neural flow of emotional visual information along the subcortical route has never been concurrently investigated in humans to definitively show that the subcortical route is a direct, afferent pathway specifically associated with fear (*Pessoa and Adolphs, 2010*; *Pessoa and Adolphs, 2011*). Such a finding would have important implications for the very foundation of visual threat perception, given this pathway's potential for rapid information transfer (*McFadyen et al., 2017*; *Silverstein and Ingvar, 2015*) and unconscious processing (*Tamietto et al., 2010*). Here, we aimed to comprehensively investigate this putative amygdala pathway in a large sample of over 600 healthy human adults from the Human Connectome Project (HCP) dataset using a multimodal imaging approach to encompass structure, function and behaviour. First, we used DWI to reconstruct the subcortical route from the superior colliculus to the amygdala, via the pulvinar, and estimated its fibre density in a large sample. Next, we modelled the direction-specific flow of haemodynamic responses to faces, testing whether a functional subcortical route is recruited to transmit information toward the amygdala. Finally, we asked whether the fibre density of the subcortical route predicts both fearful face recognition as well as the strength of dynamic coupling between the superior colliculus, pulvinar, and amygdala.

## Results

### Reconstructing the subcortical route using tractography

The first step in our investigation was to evaluate the evidence for an anatomical subcortical route to the amygdala in the healthy human brain. We exploited high-quality neuroimaging data from a large sample of 622 participants made available by the HCP (*Van Essen et al., 2013*). We then reconstructed the white matter structure of the subcortical route using two complementary tractography methods for cross-validation. We began with global tractography, a Bayesian approach to reconstructing whole-brain fibre configurations that best explain DWI data (see Materials and Methods for details). We discovered that the superior colliculus (SC) was connected to the pulvinar (PUL; fibre counts for PUL; left: $M = 13.23$, $SD = 5.56$, right: $M = 13.00$, $SD = 5.59$, minimum of 2 fibres per participant). The pulvinar and the amygdala were also connected (fibre counts for left: $M = 5.33$, $SD = 2.79$, and right: $M = 6.75$, $SD = 2.90$), with most participants having at least one connecting fibre (zero fibres for left PUL-AMG for eight participants – only 1.28% of total sample). Thus, this relatively conservative method of fibre reconstruction (as it takes into account the entire brain) can reliably detect evidence for a subcortical route across a large sample of participants.

We used the probabilistic JHU DTI-based white matter atlas (*Mori et al., 2009*), implemented in FSL, to examine any overlap between 20 major fasciculi and the globally reconstructed fibres. After warping the tractograms into standard space and converting them into track density images, we calculated the total fibre density within each fasciculus. This revealed that up to 60% of the subcortical route overlapped with major fasciculi, mainly the anterior thalamic radiation and the corticospinal tract, as well as the inferior longitudinal and fronto-occipital fasciculi. For the SC-PUL pathway, the major overlap was found in the anterior thalamic radiation in the left ($M = 56.11\%$, $SD = 15.56\%$, range = 9.46% to 100%) and right ($M = 55.78\%$, $SD = 16.97\%$, range = 4.81% to 96.55%) hemispheres, followed by the corticospinal tract (left: $M = 4.82\%$, $SD = 6.08\%$, range = 0% to 33%; right: $M = 21.06\%$, $SD = 12.75\%$, range = 0% to 69.09%; see *Figure 1*). All other fasciculi had mean track densities less than 0.06% of the full SC-PUL pathway. Track density of the PUL-AMG pathway was mostly found in the corticospinal tract (left: $M = 33.69\%$, $SD = 18.56\%$, range = 0% to 88.89%; right: $M = 36.73\%$, $SD = 17.80\%$, range = 0% to 82.35%), followed by the anterior thalamic radiation (left: $M = 20.58\%$, $SD = 14.10\%$, range = 0% to 70%; right: $M = 11.32\%$, $SD = 8.88\%$, range = 0% to 52.75%). There was also some overlap with the inferior longitudinal fasciculus (left: $M = 5.44\%$, $SD = 8.60\%$, range = 0% to 65.52%; right: $M = 3.46\%$, $SD = 5.63\%$, range = 0% to 40.68%) and the inferior fronto-occipital fasciculus (left: $M = 1.49\%$, $SD = 6.73\%$, range = 0% to 86.96%; right: $M = 7.24\%$, $SD = 10.33\%$, range = 0% to 75.72%). Mean track densities in all other fasciculi were less than 0.30%.

After covarying out head motion and removing four participants with outlying standardised residuals ($z$-score threshold ±3), we established that there were significantly more fibres connecting the SC and PUL ($M = 13.119$, *95% CI* = [12.738, 13.500]) than the PUL and AMG ($M = 6.040$, *95% CI* =

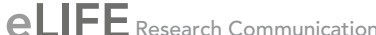

**Figure 1.** Global and local probabilistic tractography reconstructions of the subcortical route to the amygdala. Fibres reconstructed between the superior colliculus and the pulvinar are shown in (**A**) and between the pulvinar and amygdala in (**B**). Both (**A**) and the top row of (**B**) show 3D-renders of the major ROIs (amygdala in pink, pulvinar in orange, superior colliculus in green), as well as the left anterior thalamic radiation (orange) and the right corticospinal tract (pink). The reconstructed fibres for all participants are rendered by semi-transparent white streamlines. Streamlines of a single example participant are shown below in boxes. The bottom half of (**B**) shows a different, top-down perspective of the pulvinar to amygdala pathway, illustrating the right inferior frontal occipital fasciculus (light blue) and the left inferior longitudinal fasciculus (dark blue). (**C**) shows the global fibre counts (left graph) and average fibre density (right graph) for global and local tractography, respectively (SC-PUL in yellow, PUL-AMG in purple). Group average is indicated by solid white line, with dotted lines indicating the 95% confidence interval. (**D**) shows the average number of fibres terminating in each subregion of the pulvinar and amygdala, for both global and local tractography. 3D renders of the subregions (colour-coded to match the graphs)

*Figure 1 continued on next page*

*Figure 1 continued*
are shown on the left. Graphs of the number of fibres terminating in pulvinar subregions are shown for tracts connecting the superior colliculus and pulvinar (first and second graphs) and the pulvinar and amygdala (third and fourth graphs), while fibres terminating in amygdala subregions are shown for tracts connecting the pulvinar and amygdala (fifth and sixth graphs). Error bars represent 95% confidence intervals.
DOI: https://doi.org/10.7554/eLife.40766.003

[5.867, 6.214]; $F(1,616) = 433.286$, p=$2.842 \times 10^{-73}$, $\eta_p^2$ =.413). We also found a hemispheric lateralisation, such that there were more reconstructed fibres for the right (M = 9.879, *95% CI* = [9.624, 10.135]) than the left (M = 9.280, *95% CI* = [9.023, 9.537]) hemisphere ($F(1,616) = 7.583$, p=0.006, $\eta_p^2$ =.012), specifically for the PUL-AMG pathway ($F(1,616) = 16.025$, p=$7.000 \times 10^{-5}$, $\eta_p^2$ =.025; $t$(617) = −9.785, p=$4.070 \times 10^{-18}$, 95% CI [−1.714,–1.141]).

To uncover more anatomical features of the reconstructed fibres, we used subregion-specific masks of the amygdala (basolateral, centromedial, and superficial) and the pulvinar (anterior, medial, superior, inferior, and lateral; see Materials and methods for ROI specification details) to determine where the reconstructed fibres terminated. This masking approach revealed that the global tractography fibres present between the SC and PUL connected predominantly to the inferior and anterior pulvinar (see *Appendix 1—tables 3 to 5* for detailed statistics). Between the PUL and AMG, fibres terminated almost exclusively in the inferior PUL and then predominantly in the basolateral AMG. Hence, the inferior pulvinar served as the connecting node between the SC and the basolateral AMG for the globally reconstructed subcortical pathway.

To assess the validity of our findings we used a second tractography method, namely 'local' probabilistic streamline tractography, as used by *Rafal et al. (2015)* to reconstruct the subcortical route to the amygdala (*Rafal et al., 2015*). We generated streamlines between our regions of interest (ROIs) and found that the superior colliculus connected to the pulvinar (streamline counts for left: M = 1403.32, SD = 417.16, right: M = 111.59, SD = 358.60, minimum of six streamlines per participant) and the pulvinar connected to the amygdala (left: M = 575.42, SD = 203.03, right: M = 575.42, SD = 248.85, minimum 66 streamlines per participant). To evaluate whether these streamlines counts were reconstructed significantly above chance, we compared the numbers with those produced by a null distribution algorithm (*Morris et al., 2008*). We found that the number of streamlines was significantly different from chance for each connection, as determined by a series of paired two-sided $t$-tests (see *Appendix 1—table 9*), suggesting that the DWI data produced meaningful streamlines between our ROIs.

We employed a recently developed method, SIFT2, which estimates the apparent fibre density of the streamlines connecting two regions of interest. This method more accurately represents the true underlying white matter structure (*Smith et al., 2015*). The apparent fibre density of the streamlines generated using local tractography followed the same pattern as the global tractography fibre counts, such that there was greater fibre density for the SC-PUL connection (M = 5.793, *95% CI* = [5.663, 5.923]) than the PUL-AMG connection (M = 4.461, *95% CI* = [4.368, 4.554]; $F(1,607)$ = 69.586, p=$4.930 \times 10^{-16}$, $\eta_p^2$ =.103), after accounting for head motion and removing 13 outliers according to their residuals. Fibre density was also greater on the right (M = 4.935, *95% CI* = [4.822, 5.048]) than the left (M = 3.987, *95% CI* = [3.889, 4.086]) for the PUL-AMG connection ($t$(608) = −18.205, p=$1.960 \times 10^{-59}$, 95% CI [−1.050,–0.845]) while, in contrast, there was greater fibre density for the left than right SC-PUL connection ($t$(608) = 10.749, p=$8.600 \times 10^{-25}$, *95% CI* = [0.742, 1.073]; $F(1,607) = 162.475$, p=$3.828 \times 10^{-33}$, $\eta_p^2$ =.211). Taken together, our tractography analyses provide strong convergent evidence for a subcortical white matter pathway to the amygdala in the human brain.

Like in the global tractography, we investigated the overlap between the locally generated tracks and known white matter fasciculi. The pattern of results was the same, with up to 60% of fibres traversing the anterior thalamic radiation, corticospinal tract, and inferior longitudinal and fronto-occipital fasciculi. For the SC-PUL pathway, the majority of track density was found in the anterior thalamic radiation (left: M = 52.85%, SD = 11.70%, range = 16.55% to 96.96%; right: M = 58.21%, SD = 16.24%, range = 5.49% to 89.30%) and the corticospinal tract (left: M = 12.89%, SD = 8.78%, range = 0.18% to 37.84%; right: M = 32.35%, SD = 12.49%, range = 0.56% to 63.64%; see *Figure 1*). For the PUL-AMG pathway, the majority was found in the corticospinal tract (left: M = 32.29%,

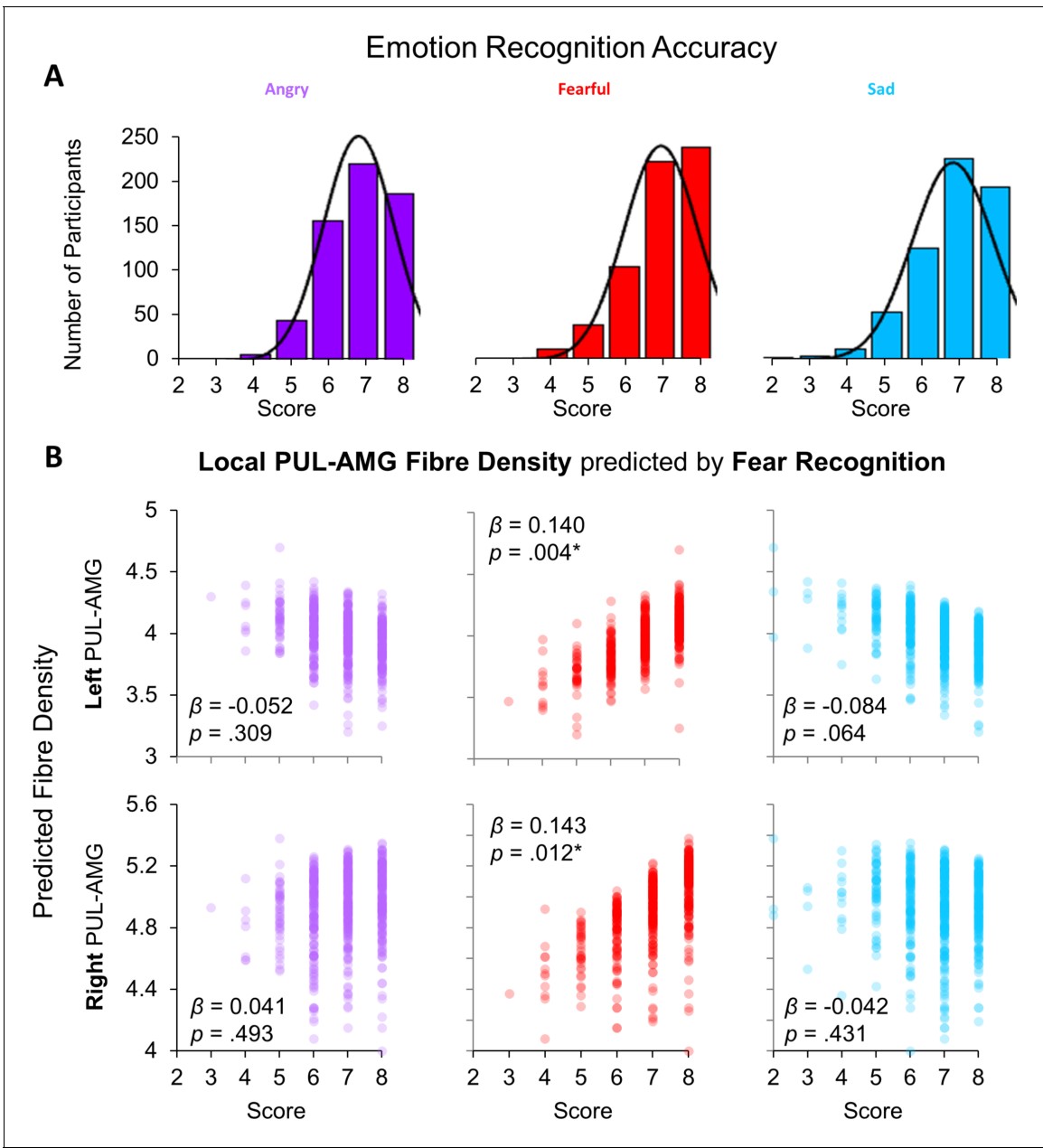

**Figure 2.** Relationship between behavioural performance and tractography. (A) Histograms of scores (out of eight) for correctly recognising the emotional expression of angry, fearful, and sad faces from the Penn Recognition Test. Normal distribution function is plotted. (B) The relationships between recognition of angry (purple), fearful (red), and sad (blue) faces (x-axes) and the predicted local fibre densities for the left (top row) and right (bottom row) PUL-AMG connection (y-axes), as produced by a multivariate ANOVA. Fearful face recognition accuracy was significantly related to local fibre density for the left and right PUL-AMG connections (β and p values shown for parameter estimates from multivariate ANOVA, *$p < .05$).

DOI: https://doi.org/10.7554/eLife.40766.004

$SD$ = 15.58%, range = 0.59% to 65.25%; right: $M$ = 37.38%, $SD$ = 16.21%, range = 0.32% to 65.99%), followed by the anterior thalamic radiation (left: $M$ = 16.47%, $SD$ = 9.20%, range = 5.24% to 59.52%; right: $M$ = 7.00%, $SD$ = 3.49%, range = 1.46% to 50.09%), and then the inferior longitudinal (left: $M$ = 5.69%, $SD$ = 7.92%, range = 0% to 50.75%; right: $M$ = 3.96%, $SD$ = 6.33%, range = 0% to 47.51%) and fronto-occipital (left: $M$ = 1.54%, $SD$ = 4.21%, range = 0% to 72.79%; right: $M$ = 7.96%, $SD$ = 10.99%, range = 0% to 79.60%) fasciculi. Mean track densities were lower than 0.20% and 0.01% in other fasciculi for SC-PUL and PUL-AMG, respectively. We also examined which subregions of the pulvinar and amygdala the seeded probabilistic tracks terminated in. For the SC-

PUL pathway, the greatest number of streamlines terminated in the anterior PUL, followed by the inferior pulvinar (see *Appendix 1—tables 6 to 8* for detailed statistics), consistent with the global tractography. Also like the global tractography, the local tractography fibres between the PUL and AMG terminated almost exclusively in the inferior PUL. For the AMG, however, fibres terminated predominantly in the basolateral subregion in the left hemisphere (consistent with the global tractography) but in the centromedial amygdala on the right.

## Greater fibre density predicts better fearful face perception

We wanted to translate our work in humans to animal research that has demonstrated clear relationships between the anatomical presence of a subcortical route and fearful behaviour (*Zhou et al., 2017*; *Wei et al., 2015*; *Shang et al., 2015*). To this end, we examined behavioural data from an out-of-scanner task, the Penn Emotion Recognition task, that assessed a different component of face processing than the in-scanner task (analysed below). In the Penn Emotion Recognition task, participants were serially presented with 40 faces that were either happy, sad, angry, fearful, or neutral (8 faces presented in each category). Participants were most accurate with identifying the emotional expression of happy faces ($M = 7.96$, $SD = 0.21$), followed by neutral ($M = 7.22$, $SD = 1.18$), and then fearful faces ($M = 7.02$, $SD = 1.03$). Recognition was poorest for angry ($M = 6.86$, $SD = 0.98$) and sad faces ($M = 6.82$, $SD = 1.12$).

We then investigated the association between these scores (see *Figure 2A*) with the fibre density of the subcortical route. We chose not to include happy or neutral expressions in our analysis because the data were substantially negatively skewed (skewness for: happy = $-5.821$; neutral = $-2.053$; angry = $-0.719$; fearful = $-1.188$; sad = $-1.090$). Thus, we entered fibre density measures for the SC-PUL and PUL-AMG pathways into two separate multivariate regressions (one per tractography method, to reduce collinearity) with recognition accuracy scores for fearful, angry, and sad faces as covariates, plus head motion as a control covariate. We removed outliers (four for global tractography, 15 for local tractography) with z-scored residuals $\pm 3$. While there were no significant multivariate relationships between global tractography and emotion recognition (see *Appendix 1—tables 10* and *11* for detailed statistics), there was a significant relationship between local tractography and recognition accuracy for fearful faces ($F(4,598) = 2.501$, p=0.042, Wilk's $\Lambda = 0.984$, $n_p^2 = 0.016$; see *Figure 2B*). This was driven predominantly by fibre density of the left ($\beta = 0.140$, p=0.004) and right ($\beta = 0.143$, p=0.012) PUL-AMG connections' local fibre density. The local fibre

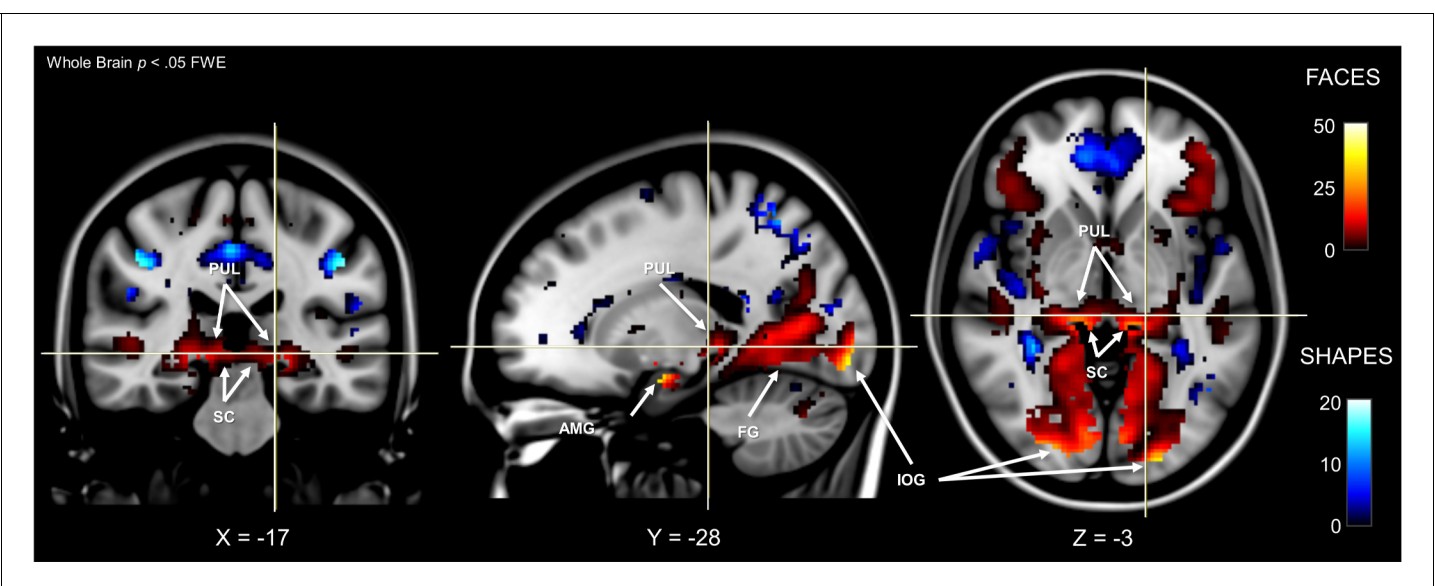

**Figure 3.** Whole-brain fMRI activation for faces and shapes. Face activation is shown by hot colours and shape activation is shown by cool colours. MNI coordinates are shown at the bottom. Labels indicate significant (p < 0.05, whole brain FWE) activation in the superior colliculus (SC), pulvinar (PUL), amygdala (AMG), inferior occipital gyrus (IOG), and fusiform gyrus (FG). Results are overlaid on an averaged MNI152 T1 template.
DOI: https://doi.org/10.7554/eLife.40766.005

density of the left and right SC-PUL did not contribute significantly to the model. These results suggest that the fibre density of the PUL-AMG half of the subcortical route is associated with fearful face recognition more so than with other negative (sad) or threatening (angry) emotional expressions.

## Subcortical and cortical BOLD signal to emotional faces

We used dynamic causal modelling to infer the dynamic (or effective) connectivity between each node of the subcortical route and determine the directionality of the functional interactions occurring along the anatomical pathway mapping described above. First, it was necessary to establish any differences in functional activation within these nodes. To do this, we used the 'Emotion' task from the HCP battery of fMRI experiments, in which participants performed a matching task using images of faces or shapes. We contrasted activation in face versus shape blocks and reviewed the results at the whole-brain group level across all 622 participants, p<0.05 FWE (see *Figure 3*). This revealed a network of significant BOLD clusters spread across occipital, temporal, frontal, parietal, and subcortical areas, replicating previous work with this dataset (*Barch et al., 2013*). Critically, the most significant cluster included the left and right amygdala as well as the left and right fusiform gyri (FG) and inferior occipital gyri (IOG). These latter two regions are key nodes in the cortical visual processing stream for faces, which may feed information forward to the amygdala (*Tamietto et al., 2010*).

We used the SPM Anatomy Toolbox to confirm the anatomical positions of our functional activations. In the absence of an anatomical template for the superior colliculus and the pulvinar, we masked the map of statistically significant voxels (p < 0.05, FWE) with our a priori manual anatomically defined superior colliculi mask and functionally-defined pulvinar masks from *Barron et al. (2015)*; see Materials and Methods for ROI generation). This revealed significant voxels in each area (proportion of significant voxels within each mask: left SC = 65.08%, right SC = 73.55%, left PUL = 36.13%, right PUL = 51.49%). Therefore, the faces-vs-shapes fMRI HCP task established functional activation in the three subcortical nodes of interest, as well as two nodes of a potential cortical pathway to the amygdala for conveying information about emotional faces.

## A forwards-only subcortical route is engaged during face processing

After observing significant BOLD signal in regions along the subcortical route as well as in other visual cortical areas, we next asked whether these regions were dynamically connected. We designed a space of testable models that mapped onto the functional hypothesis of a subcortical route to the amygdala that operates alongside a cortical visual pathway and is modulated by faces. Due to the presence of the IOG and FG in the whole-brain corrected fMRI activation and their known roles in face processing (*Johnson, 2005*), we defined several plausible functional cortical connections to the amygdala. These consisted of reciprocal pathways between IOG and FG, FG and amygdala, IOG and amygdala, as well as pulvinar and IOG (see *Figure 4*). Note that, while previous research has defined motion-related area V5/MT as a significant component of the pulvinar's subcortical visual network (*Zhou et al., 2017*), we did not observe strong involvement of this area in the faces-vs-shapes fMRI task (12% probability in cluster 37 with two voxels). Hence, we omitted area V5/MT from our model space. We named models containing both a cortical and subcortical route to the amygdala as 'Dual' models, whereas models in which the subcortical route was absent were named 'Cortical'. Our model space also included different sources of visual input, namely to the superior colliculus, pulvinar, or both, given that the pulvinar also receives direct retinal input (*Cowey et al., 1994*) as well as input via the superior colliculus (*Berman and Wurtz, 2011*). This gave us six families of models: 1) Cortical with SC input, 2) Cortical with PUL input, 3) Cortical with SC and PUL input, 4) Dual with SC input, 5) Dual with PUL input, and 6) Dual with SC and PUL input. We considered all possible combinations of forwards and reciprocal (forwards and backwards) cortical and subcortical connections, giving us a comprehensive model space of 102 models.

Of the available 622 participants, we conducted dynamic causal modelling on a subset of 237 participants who had sufficient above-threshold activation in all ROIs (these were defined by the subcortical masks used thus far and by spheres surrounding the coordinate of peak group BOLD signal in the IOG and FG; see Materials and Methods for more details). We conducted Bayesian model selection on the model space (grouped by families) to estimate how well the models explained the data, taking into account model complexity. We used the random effects implementation to account for

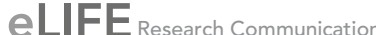

**Figure 4.** Dynamic causal modelling model space and estimated probabilities. (A) Diagram of the model space constructed for the fMRI activation to viewing faces. The top row shows various types of model designs in the Cortical family, and the bottom row shows model designs in the Dual family (which includes a subcortical route). Each model variation included different combinations of forwards and backwards connections, indicated by dashed arrows. The model numbers are shown above each model variation. Every model contained input to the inferior occipital gyrus but some models also contained input to the superior colliculus (green) only, pulvinar (orange) only, or both (black). (B) The expected (top row) probabilities and exceedance probabilities (bottom row) for each of the 102 models. The colour of the bars indicates the different type of input for that family (i.e., SC, PUL, or both), according to A. Individual model probabilities are shown on the left (including a diagram of the winning model) and the probabilities for the families are shown on the right.

DOI: https://doi.org/10.7554/eLife.40766.006

potential individual differences in the recruitment of a subcortical pathway for viewing faces (*Stephan et al., 2009*). The winning family was the 'Dual with SC and PUL input' family (expected probability = 67.34%, exceedance probability = 100%) and the winning model across the entire model space was also within this family (expected probability = 21.24%, exceedance probability = 98.01%, protected exceedance probability = 98.18%; see *Figure 4*). This model included

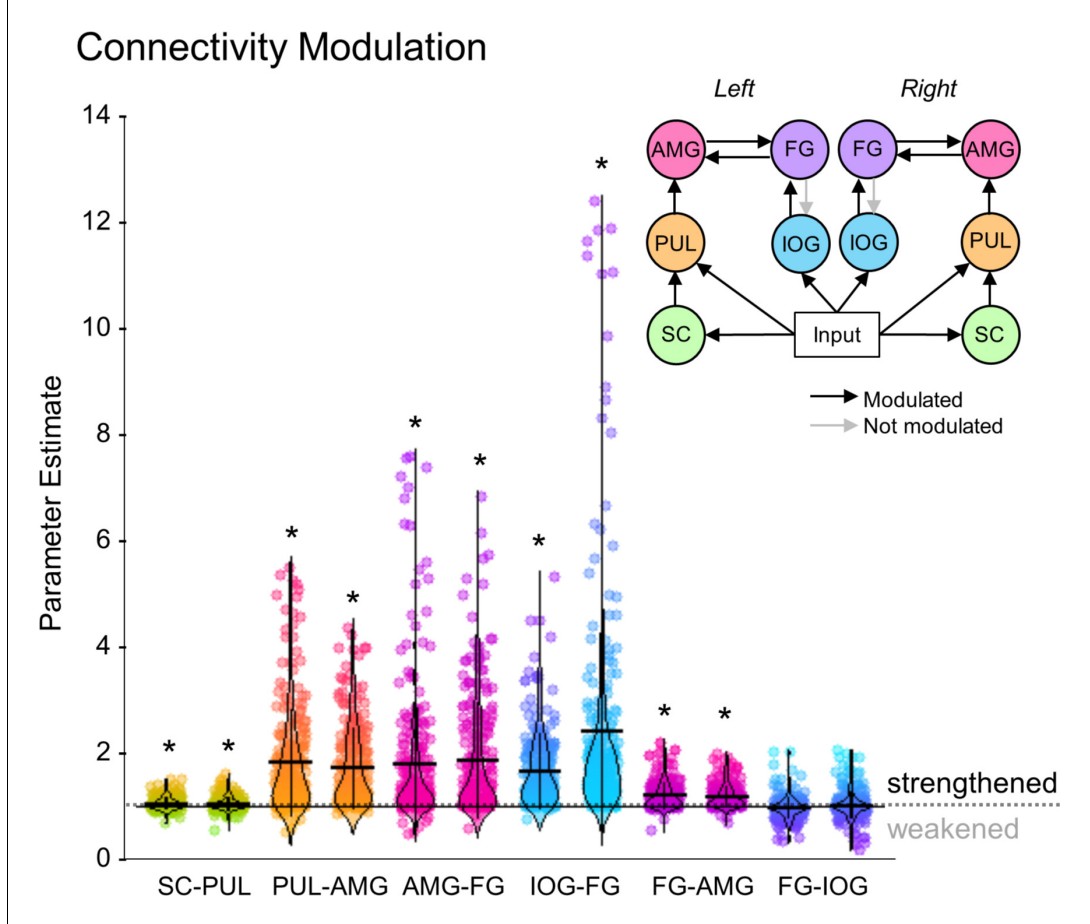

**Figure 5.** The strength of individual connections from the winning dynamic causal model. Parameter estimates of the modulatory connection strength of the winning model. Dot plots of up to 234 participants are shown with horizontal lines for the mean and vertical lines for one standard deviation. Density distribution is represented by each violin plot. The long horizontal dotted line across the graph represents the prior (set to 1), where * indicates the connection was significantly greater than 1 (p<0.05, corrected for multiple comparisons). A diagram of the winning model (left and right hemispheres are shown) is at the top right. Greyed-out connections indicate those that were not significant.
DOI: https://doi.org/10.7554/eLife.40766.007

reciprocal cortical connections between IOG and FG and between FG and the amygdala. It also included a forwards-only subcortical route from the superior colliculus to the pulvinar to the amygdala, with input to both the superior colliculus and the pulvinar. The Bayesian Omnibus Risk score was p=$1.78\times10^{-124}$, indicating a very small chance that the winning model was indistinguishable from all models tested (*Rigoux et al., 2014*). We replicated this finding (same winning family and winning model) on a subsample consisting of only the unrelated (i.e. non-sibling) participants within this group (49 participants; see Appendix 1).

The winning model revealed that the functional network that best explained the BOLD responses in our sample of 237 participants included visual inputs to the superior colliculus and pulvinar, forward connections from superior colliculus to the amygdala via the pulvinar, and recurrent interactions between IOG and FG, as well as between FG and amygdala. To extrapolate this finding to the general population and assess the consistency of dynamic coupling at each individual connection, we performed inferential statistics (*t*-tests) on the parameter estimates of each connection within the winning model (i.e. connectivity strength, in their natural space). We looked at the connectivity modulation parameters that represent the change in connection strength caused by the effect of faces. We removed extreme outliers (>3 SDs from mean) participants from each connection (*M* = 5.25, range = 3 to 8 participants excluded from sample of 237) and found that all connectivity modulations were significant (one sample *t*-tests against a test value of 1; see *Appendix 1—table 13* for detailed

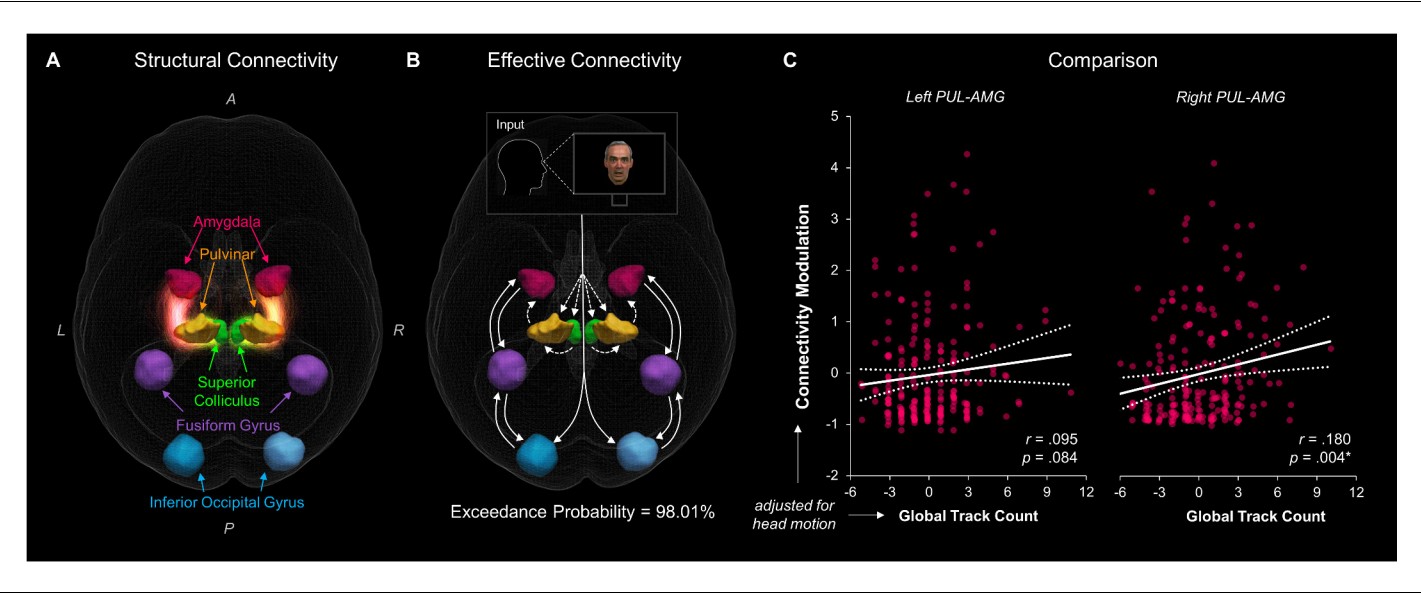

**Figure 6.** The relationship between structural and effective connectivity for the subcortical route. (**A and B**) show 3D renders of the ROIs used in the dynamic causal modelling stage. A = anterior, p = posterior, R = right, L = left. (**A**) 3D-rendered tracks generated by global tractography overlaid for all 622 participants (pink/orange for PUL-AMG, green/yellow for SC-PUL). (**B**) Direction of information flow according to the winning dynamic causal model (exceedance probability = 98.01%) illustrated using an axial view of the network. Dotted lines indicate subcortical input/connections. (**C**) Positive correlation between global track count for the left and right PUL-AMG connections and the modulatory strength of the same connection in the winning effective connectivity model (with 95% CI shown).

DOI: https://doi.org/10.7554/eLife.40766.008

statistics) except for the backward connection from left and right FG to left IOG (see *Figure 5*). These results suggest that the modulation of these connections by faces was consistently strong and so we can infer that a subcortical route for processing faces is likely present within the general population.

## Greater fibre density relates to stronger effective connectivity

Our findings from tractography, fMRI, and dynamic causal modelling provide convergent evidence for a subcortical route to the amygdala in humans. The final question we set out to answer was whether this converging evidence was correlated, such that participants with stronger structural connectivity also had stronger effective connectivity. In other words, we asked whether the structural connectivity along the subcortical amygdala route enables functional interactions amongst the nodes that lie within it. We computed eight partial correlations (with head motion as a control covariate) to examine the relationship between each parameter estimate and the corresponding global fibre count and local summed weights per connection (left and right SC-PUL and PUL-AMG). After removing multivariate outliers (leaving N = 213; see *Appendix 1—table 12* for details), we discovered that participants with more global fibres also had greater modulatory activity for the right ($r = 0.180$, p = 0.004, $p_{bonf} = 0.032$; see *Figure 6*) but not the left ($r = 0.095$, p = 0.084, $p_{bonf} = 0.672$) PUL-AMG connection. The SC-PUL connection was not significantly related to its corresponding DCM parameters for global (left: $r = -0.022$, p = 0.627, $p_{bonf} = 1.000$; right: $r = 0.002$, p = 0.488, $p_{bonf} = 1.000$) or local (left: $r = -0.101$, p = 0.928, $p_{bonf} = 1.000$; right: $r = -0.028$, p = 0.659, $p_{bonf} = 1.000$) tractography. Note that we successfully replicated this finding within a subsample of unrelated participants (49 participants; see Appendix 1). Thus, our study is the first to successfully harmonise functional and structural information about the subcortical pulvinar connection to the amygdala.

## Discussion

The elusive subcortical route to the amygdala has posed a unique challenge in studies of the human brain, due to its depth and its fast activation. Evidence has accumulated over recent years across many studies using various neuroimaging modalities showing that this pathway may underlie primitive threat-related behaviour. These studies, however, often take a unimodal approach on typically small samples, making it difficult to relate the specific structural connections between superior colliculus, pulvinar, and amygdala to observed functional brain activity and behavioural output. Our study, which used a large sample of participants from the HCP, supports the existence of a subcortical pulvinar connection to the amygdala in the healthy human adult brain that facilitates dynamic coupling between these regions and also enhances fear recognition. We reconstructed the subcortical route to the amygdala using sophisticated tractography methods and found that the white matter fibre density of the pulvinar-amygdala connection significantly predicted individuals' ability to recognise fearful faces. We then computationally modelled the functional neural networks along this structurally connected network that were engaged while people viewed emotional faces. We found that it was more likely for the network to include a subcortical visual route to the amygdala than a cortical route alone. Finally, we revealed converging evidence from structural and effectivity connectivity, such that the fibre density of the right pulvinar to amygdala pathway was positively correlated with the strength of the dynamic coupling (i.e. effective connectivity) between these regions.

This study marks the first time that structural and effective connectivity have been concurrently investigated in the one large sample to address the controversy on the existence and functional role of the putative subcortical route to the amygdala. Up to 60% of its fibre density overlapped with major fasciculi, including the corticospinal tract, anterior thalamic radiation, inferior longitudinal fasciculus, and inferior fronto-occipital fasciculus. Tractography of diffusion images is susceptible to both false positives and false negatives and thus is seldom used in isolation to determine the existence of particular neuroanatomical pathways (*Jbabdi and Johansen-Berg, 2011*). We established the validity of our tractographically reconstructed subcortical route by directly relating our measures of fibre density to both behaviour and effective connectivity, as well as by using two different tractography methods. Had the fibre density measures been simply due to noise, we would not have expected these theoretically relevant relationships with fearful face processing to emerge within this large sample of individuals. Notably, these intermodal relationships were only found for the pulvinar-amygdala connection, despite there being greater fibre density between the superior colliculus and the pulvinar and this connection being present in the winning dynamic causal model. One explanation for this is that we had relatively less BOLD signal-to-noise ratio in the superior colliculus due to its small size and proximity to major blood vessels in the brain stem (*Wall et al., 2009*), thus weakening the likelihood of finding consistent covariance of its functional coupling with fibre density. Another explanation, particularly regarding the behaviour-tractography relationship, is that the pulvinar plays a significant functional role in the subcortical route to the amygdala. Research on macaques has demonstrated the pulvinar's response to emotional faces (*Soares et al., 2017*; *Maior et al., 2010*) and its role in modulating attention (*Soares et al., 2017*) and so we would indeed expect the strength of the pulvinar-amygdala connection to be more predictive of fearful face recognition. Future research could more deeply investigate the relative contribution of each half of the subcortical route to emotional face processing by using an optimised fMRI approach (*Wall et al., 2009*) and contrasting different types of stimuli – for example, low vs. high spatial frequency (*Gomes et al., 2017*) or moving stimuli (*Berman and Wurtz, 2011*).

Our decision to reconstruct the two halves of the subcortical route separately was motivated by our interest in the relative contribution of each connection to face-related processing (as described above) but was also a limitation imposed by anatomically-constrained tractography, where reconstructed fibres are terminated at boundaries between grey and white matter (*Smith et al., 2012*). Given that the pulvinar is made up of thalamic cell bodies (grey matter), the likelihood of reconstructing a continuous streamline of axon bundles *traversing* the pulvinar's grey matter may have been restricted by these boundary constraints. Previous studies that have not imposed these constraints have successfully traced a continuous pathway from the superior colliculus to the amygdala via the pulvinar (*Rafal et al., 2015*; *Tamietto et al., 2012*), supporting animal research showing that inferior-lateral pulvinar neurons receiving superior colliculus afferents also have efferent connections to the lateral amygdala (*Day-Brown et al., 2010*). Our investigation into pulvinar and amygdala

subregions support these findings, such that we found the superior colliculus to project predominantly onto the inferior (and anterior) pulvinar, which was the same subregion to receive the vast majority of fibres from the amygdala (see *Figure 1*). Furthermore, pulvinar fibres terminated predominantly within the basolateral amygdala, which is known to process visual information about threat and faces (*Hortensius et al., 2016*). Further studies could use both anatomically constrained tractography and this subregion-specific approach with ultra-high-resolution imaging to better differentiate grey-white matter boundaries and more accurately determine if and where a continuous, subcortical route might traverse the pulvinar.

While our results suggest that the inferior pulvinar may serve as a disynaptic connection point between the superior colliculus and amygdala, the continuity of information flow along the subcortical route is still a disputed feature due to the strong cortical influences on the pulvinar (*Bridge et al., 2016*; *Pessoa and Adolphs, 2011*). This dispute has also arisen from prior work investigating the spatial frequency content of information conveyed along the subcortical route. Research on blindsight patients has found evidence only for low spatial frequencies which suggests that such information originated from magnocellular cells in the superior colliculus (*Burra et al., 2017*; *Méndez-Bértolo et al., 2016*). On the other hand, work in healthy participants has found no such spatial frequency preference, which suggests that rapid pulvinar-amygdala transmission might include input from other parvocellular pathways (*McFadyen et al., 2017*). We did not exhaustively explore the extent to which the cortex contributes information to the pulvinar-amygdala connection. The winning effective connectivity model, however, did not include cortical connections between the pulvinar and the inferior occipital gyrus. Hence, it is unlikely that the primary visual cortex contributed (either via direct anatomical connections or functional coupling along the ventral visual stream; *Pessoa and Adolphs, 2010*) to the information transmitted along the subcortical route. The winning model did, however, include input to the superior colliculus as well as directly to the pulvinar, which could reflect direct retinal input or input from areas not explicitly included in the model, such as the parietal cortex, temporal cortex, or the LGN (*Bridge et al., 2016*), that may transmit both low and high spatial frequency information. Furthermore, it remains to be shown how interactions between the pulvinar and other cortical areas, such as the inferotemporal cortex (*Zhou et al., 2016*), may directly influence activity along the pulvinar-amygdala connection.

Our findings open avenues for future studies on how this subcortical pathway might influence threat-related behaviour. While our findings demonstrated that greater pulvinar-amygdala fibre density related to better fearful face recognition, it remains to be seen how this might compare with structural connectivity of other cortical networks. In other words, would the fibre density of this subcortical connection explain fearful face recognition above and beyond, say, structural connections between the inferior temporal or orbitofrontal cortex and the amygdala (*Pessoa and Adolphs, 2011*) or between the thalamus and the superior temporal sulcus (*Leppänen and Nelson, 2009*)? Evidence from blindsight patients suggests that this subcortical connection ensures redundancy and compensation, such that it strengthens when cortical connections are destroyed (*Tamietto et al., 2012*). Taking this in conjunction with our findings, we might consider that the pulvinar-amygdala connection contributes to fear recognition in faces (and effective connectivity underlying face perception) in healthy participants but can increase or decrease its influence depending on the functioning of other networks. Such increases and decreases are already evident in certain clinical populations. For example, structural connectivity between the superior colliculus, pulvinar, and amygdala is weakened in individuals with autism compared to healthy controls (*Hu et al., 2017*), and BOLD signal to fearful faces is reduced in these areas (*Kleinhans et al., 2011*; *Green et al., 2017*), unless participants are explicitly instructed to fixate on the eyes (*Hadjikhani et al., 2017*). On the other hand, people who suffer from anxiety show hyperactive activity along the subcortical route compared to non-anxious individuals (*Hakamata et al., 2016*; *Tadayonnejad et al., 2016*; *Nakataki et al., 2017*). How and why this subcortical visual pathway to the amygdala is altered in these clinical populations remains a significant and relatively unexplored avenue of research.

We observed hemispheric lateralisation of the pulvinar-amygdala connection, such that both the local and global tractography showed greater fibre density along the right than the left, and there were stronger tractography-behaviour and tractography-connectivity relationships for the right than the left. Early studies on the subcortical route observed specifically right-sided BOLD responses during non-conscious fearful face viewing (*Morris et al., 1999*; *Morris et al., 1998*), and a previous tractography study has also found that only the fractional anisotropy of the right subcortical route was

significantly related to threat-biased saccades (*Koller et al., 2018*). There is mounting evidence for right-sided specialisation for ordered (*Wyczesany et al., 2018*) and disordered (*McDonald, 2017*) emotion processing, particularly for non-conscious signals transmitted along the subcortical route (*Gainotti, 2012*). Thus, our results lend support to this theory by demonstrating evidence for the right pulvinar-amygdala connection's stronger fibre density and its relationship to emotional face viewing and fearful face recognition. Our understanding of this lateralisation may be deepened by future exploration of left- vs. right-sided structural connectivity and function along the subcortical route during conscious vs. non-conscious emotion processing in healthy participants.

One limitation of the present study is the discrepancy between how local and global measures of fibre density related to other measures; namely, that local tractography covaried with fearful face recognition scores while global tractography covaried with effective connectivity. While the reconstructed fibres shared many similarities (e.g. the pattern of findings for each connection across hemispheres and subregions, as well as the overlap with major fasciculi; see *Figure 1*) even after accounting for head motion, it is possible that the local tractography's relatively greater susceptibility to noise may have decreased its relationship to corresponding effective connectivity parameters. Indeed, global tractography has been shown to better reflect local connection architecture (*Jbabdi and Johansen-Berg, 2011*), such as the subcortical connections we have investigated. Such discrepancies between global and local tractography have been reported in other work (*Anastasopoulos et al., 2014*) and so further research (particularly those that only recruit a single tractograpy method) will benefit from specific investigations into why these discrepancies might arise.

In conclusion, our study has made substantial progress towards settling the long-held debate over the existence and function of a subcortical route to the amygdala in the human brain. Our multimodal neuroimaging approach, leveraged by computational modelling, provides convergent evidence for a fundamental and conserved pulvinar-amygdala pathway that is specifically involved in fear. We demonstrate that the white matter tracts that form the subcortical structural pathway from the pulvinar to the amygdala enables functional, dynamic interactions involved in emotional face perception. Critically, we show that structural connectivity between the pulvinar and the amygdala leads to better recognition of fearful expressions.

## Materials and methods

### Participants

We used the data from the publicly available Human Connectome Project (HCP) S900 release, collected between 2012 and 2016, containing data from 897 consenting adults (*Van Essen et al., 2013*). Ethical permission to use this data and the associated restricted access data (including variables such as specific age information) was obtained from the University of Queensland Human Research Ethics Committee. Out of these participants, 730 young adults had complete MRI and dMRI data, as well as fMRI data for the faces-vs-shapes task (*Van Essen et al., 2012*). Of these, we excluded 95 people due to positive drug/alcohol tests and an additional 13 for abnormal colour vision. This resulted in a final sample of 622 participants aged between 22 and 36 years (M = 28.81, SD = 3.68 years), 259 of whom were male and 363 female, with 569 right-handed and 53 left-handed. Within our sample, 495 participants were related to one or more other participants (328 families in total). This included 53 pairs of monozygotic twins, 50 pairs of dizygotic twins, and 289 participants with one or more non-twin siblings in the sample. The remaining 127 participants were unrelated. We acknowledged that the many siblings in the HCP sample might spuriously decrease the variance in our neural measures (due to the structural and functional similarity between siblings, for example) and thus influence our statistics. Because of this, we replicated some of the analyses from the full sample on the subsample of unrelated participants (see Appendix 1).

### dMRI processing
#### dMRI acquisition
The HCP scanned participants in sessions over 2 days using a custom-made Siemens 3T 'Connectome Skyra' (Siemens, Erlangen, Germany) with a 32-channel head coil, located at Washington University, St Louis, USA. They collected two separate T1-weighted high-resolution MPRAGE images

(voxel size = 0.7 mm isotropic, field of view = 224 mm, matrix = 320, 256 saggital slices, TR = 2,400 ms, TE = 2.14 ms, TI = 1,000 ms, flip angle = 8°, bandwidth = 210 Hz per pixel, echo spacing = 7.6 ms). We only used the first T1 image of the two sessions in our analysis. The HCP collected multi-band multi-shell diffusion-weighted images in a single session also using the Connectome Skyra (three shells with b-values of 1000, 2000, and 3000 s/mm$^2$; 90 directions per shell; voxel size = 1.25 mm isotropic; TR = 5520 ms; TE = 89.5 ms; flip angle = 78°; field of view = 210×180 mm; refocusing flip angle = 160°; echo spacing = 0.78 ms; bandwidth = 1488 Hz/pixel; slice thickness = 111×1.25 mm).

### dMRI preprocessing

We used the minimally processed images provided by the HCP. For the T1 images, this included gradient distortion correction, bias field correction (using FSL: Jenkinson, Beckmann, Behrens, Wool-rich, and Smith, 2012; http://fsl.fmrib.ox.ac.uk/fsl/fslwiki/), and cortical segmentation (using FreeSur-fer: Dale, Fischl, and Sereno, 1999; http://surfer.nmr.mgh.harvard.edu/). For the diffusion images, this included intensity normalisation across runs, echo planar imaging (EPI) distortion correction and eddy current/motion correction (for full details, see *Glasser et al., 2013*). The latter stage produced motion parameters (three translations and three rotations), across which we computed the mean rel-ative displacement between dMRI volumes. These values per participant were used in all further analyses to account for any confounding effect of motion (*Baum et al., 2018*). We conducted all fur-ther processing using MRTrix 3.0.15 (*Tournier et al., 2012*) and FSL 5.0.7.

### Global intensity normalisation

First, we corrected for low-frequency B1 field inhomogeneities in the dMRI volumes. We then con-ducted global intensity normalisation across each participant's corrected diffusion-weighted image so that we could later perform quantitative analyses of fibre density (i.e. apparent fibre density; *Raffelt et al., 2012*). This step normalises the median white matter b = 0 intensity (i.e. non-diffusion-weighted image) across participants so that the proportion of one tissue type within a voxel does not influence the diffusion-weighted signal in another. Given our large sample size, we selected a subset of 62 participants (approximately 10% of the sample) to create a representative fractional anisotropy (FA) population template and white matter mask. We then used the population template and white matter mask to normalise the white matter intensity of all 622 participants' dMRI volumes.

### Response function estimation

We segmented each participant's T1 image into five tissue types (cortical grey matter, subcortical grey matter, white matter, CSF, and pathological tissue) using the Freesurfer parcellation image pro-vided by the HCP. We then estimated response functions (i.e. the signal expected for a voxel con-taining a single, coherently-oriented fibre bundle) for grey matter, white matter, and CSF using the Multi-Shell Multi-Tissue (MSMT) Constrained Spherical Deconvolution (CSD) algorithm (*Jeurissen et al., 2014*). After completing this step for all participants, we averaged their response functions to produce representative response functions per tissue type. We then conducted MSMT CSD on each participant again using the group averaged response functions, producing individual multi-tissue fibre orientation distributions (FODs).

## fMRI processing

### fMRI acquisition

As with the dMRI data, the HCP acquired whole-brain gradient-echo echo planar imaging (EPI) data using the Connectome Skyra (TR = 720 ms, TE = 33.1 ms, flip angle = 52°, bandwidth = 2,290 Hz/Px, in-plane field of view = 208×180 mm, 72 slices at 2 mm thick, voxel size = 2 mm isotropic, echo spacing = 0.58 ms) with a multiband factor of eight. They collected data in a one-hour session (either on the same day as the dMRI or one day before/after) along with two or three other functional tasks in the HCP battery. For the faces-vs-shapes task, there were two runs, one with right-to-left phase encoding and the other with left-to-right phase encoding, each with 176 frames at a duration of 2 min and 16 s.

### fMRI task

The HCP developed the 'emotion task' (i.e. faces vs. shapes) from the paradigm presented by *Hariri et al. (2002)*. Participants were presented with three visual stimuli at a time (one image at the top and two at the bottom) using E-Prime (*Schneider et al., 2002*). Participants were then instructed to make a button press indicating which of the two images at the bottom (left or right) matched the image at the top. Images were either a face (angry or fearful) or a shape (circle, horizontal oval, or vertical oval). The stimuli were presented on screen for 2000 ms separated by a 1000 ms inter-stimulus interval. A cue was presented at the beginning of each block to indicate the block type (i.e. 'shape' or 'face'), where each block contained either six faces trials or six shapes trials. Finally, a fixation cross was presented for eight seconds at the end of each of each run. The last block of each run only contained the first three trials due to a technical error that occurred early in HCP data collection. As the first block was always a shape block, our analysis was conducted on three shape blocks and 2.5 face blocks.

### fMRI preprocessing

We used the minimally preprocessed fMRI data provided by the HCP corrected for gradient distortion, motion, and field map-based EPI distortion. The HCP intensity normalised the data and spatially transformed it to MNI152 space using FSL (see *Glasser et al., 2013*) for full details on preprocessing pipeline). We further increased the signal-to-noise ratio of the fMRI data in SPM12 (SPM12, www.fil.ion.ucl.ac.uk/spm) by applying spatial smoothing using a 4 mm Gaussian kernel (*Hillebrandt et al., 2014*).

## Regions of interest

We chose the superior colliculus, pulvinar, and amygdala as our ROIs. We created masks of these ROIs in standard MNI space using FSL. For the amygdala (AMG) binary mask, we used the probabilistic Harvard-Oxford Subcortical atlas at a threshold of at least 50% probability. For amygdala subregions, we used the basolateral, centromedial, and superficial amygdala regions in the Juelich Histological Atlas (*Amunts et al., 2005*) at a threshold of at least 40% probability. For the pulvinar (PUL), we were interested in the structure as a whole, as well as its subregions (results for the latter are detailed in Appendix 1). To do this, we used the parcellated pulvinar mask generated by *Barron et al. (2015)*, who isolated five distinct pulvinar clusters based on functional co-activation profiles in fMRI data from 29,597 participants across 7772 experiments (*Barron et al., 2015*). For the pulvinar as a whole ROI, we merged the five clusters together and used FSL to manually fill any holes in the resultant binary mask. Finally, we manually created binary masks for the left and right superior colliculi (SC) in the absence of an atlas-based mask by drawing the boundaries of the superior colliculus over the MNI152 single participant T1 template with reference to an anatomical atlas (*Tamraz and Comair, 2004*) and filling the centre. We then used FSL to warp these masks into native diffusion space for each participant's tractography analysis. All our ROIs in MNI space are freely available online from the Open Science Framework: doi:10.17605/OSF.IO/KBPWM.

## dMRI analysis

In this study, we implemented two tractography methods that use different approaches to white matter reconstruction for cross-method validation. We first used the multi-tissue model of global tractography. This method takes a Bayesian approach to reconstructing a full-brain fibre configuration using a generative signal model to best explain the underlying data. It is less sensitive to noise that may accumulate for longer distance tracts in other 'local' tractography methods throughout their stepwise approach (*Christiaens et al., 2015*; *Reisert et al., 2011*). Hence, for comparison, we computed probabilistic ('local') tractography between our regions of interest (*Tournier et al., 2010*). This method also uses a Bayesian approach to account for one or more distributions of fibre orientations within each voxel, thus incorporating uncertainty into the model (*Zhou et al., 2017*). To acquire a biologically accurate measure of apparent fibre density (*Raffelt et al., 2012*) along the resultant streamlines, we used the Spherical-Deconvolution Informed Filtering of Tractograms version 2 (SIFT2) method to weight each streamline by a cross-sectional area multiplier directly related to the underlying data (*Smith et al., 2015*; *Raffelt et al., 2012*). For both the global (producing 'fibre count' as a variable) and local tractography with SIFT2 (producing 'summed weights' as a variable),

we computed 2 (hemisphere: left, right) by 2 (connection: SC-PUL, PUL-AMG) repeated-measures ANOVAs to quantitatively examine the properties of these pathways.

## Global tractography

Global tractography is a data-driven Bayesian approach to estimating the whole-brain fibre configuration that best explains the underlying diffusion-weighted images. As opposed to local streamline tracking, global tractography accounts for the spatial continuity of fibres and thus is better able to discriminate crossing and fanning fibre geometries (*Christiaens et al., 2015*). Furthermore, because the simultaneously-reconstructed fibre configurations are optimised with respect to the data at hand, the density of the final tractogram quantitatively represents the apparent fibre density (AFD; i.e. the proportion of space occupied by white matter fibres (*Raffelt et al., 2012*).

We conducted global tractography on the global-intensity-normalised DWI volumes for each participant using the group-averaged multi-tissue response functions. After 250 million iterations to optimise a full brain reconstruction, we filtered the tractogram using the ROI masks described above to isolate fibres that terminated in 1) both the superior colliculus and pulvinar masks, and 2) both the pulvinar and amygdala masks. We also used the masks for the five individual functionally-defined pulvinar subregions to isolate the subregion-specific fibres connecting to the superior colliculus and to the amygdala.

## Local tractography

As a less conservative approach than global tractography, we also conducted local probabilistic tractography using our ROIs as seeding and terminating regions. We used the iFOD2 algorithm, iteratively planting a seed point 25,000 times (or until at least 10,000 tracks had been selected and written) in each voxel of the seeding ROI (*Tournier et al., 2012*). We applied the anatomically-constrained variation of this technique, whereby each participant's five-tissue-type segmented T1 image provided biologically realistic priors for streamline generation, reducing the likelihood of false positives (*Smith et al., 2012*). We edited the final streamlines so that only those that terminated at white-grey matter boundaries in our ROIs remained.

Using these methods, we traced streamlines between the two halves of the subcortical route (i.e. SC to PUL, PUL to AMG). We then reversed the seeding location (i.e. PUL to SC, AMG to PUL) and based all statistics on the average between the forwards and backwards seeding directions to reduce any influence of possible asymmetries in seed ROI volume. We applied SIFT2 to these streamlines to enable us to quantitatively assess the connectivity. SIFT2 makes this possible by weighting the streamlines by a cross-sectional multiplier such that the sum of these weighting factors better represents the underlying white matter fibre density (*Smith et al., 2012*).

## fMRI analysis

### General linear modelling

Using the spatially smoothed fMRI data, we convolved the onset of each Face and Shape block with a canonical hemodynamic response function (HRF) using SPM12. We closely modelled this first-level general linear model (GLM) analysis on the work by *Hillebrandt et al. (2014)*, such that we did not slice time correct the multiband data due to the fast TR. We partitioned the GLM into sessions (left-to-right and right-to-left encoding) and we included 12 head motion parameters as multiple regressors (six estimates from rigid-body transformation, and their temporal derivatives). We generated statistical parametric maps (SPMs) of the expected BOLD signal for faces minus shapes and shapes minus faces.

We then entered the faces minus shapes contrast into a second-level analysis (a one-sample *t*-test) across all participants. After examining the estimated BOLD signal to Faces at the whole-brain level (p<0.05, family-wise error corrected), we applied the superior colliculus, pulvinar, and amygdala *a priori* defined masks to more specifically estimate functional activation in these anatomically-defined areas.

### Dynamic causal modelling

We implemented Dynamic Causal Modelling (DCM) to infer the causal direction of information flow between neural regions using a biophysically informed generative model (*Friston et al., 2003*). First,

we examined the map of significant activation produced by the fMRI analysis of the 'Emotion' (i.e. faces vs. shapes) HCP task. Based on this and our a priori hypotheses, we defined the left and right superior colliculus, pulvinar, amygdala, inferior occipital gyrus (IOG), and fusiform gyrus (FG) as our ROIs. For the two gyri, we used MNI coordinates of the most significant peak from the group level analysis (left IOG: −22–92 −10, right IOG: 28–90 −8, left FG: −38–50 −20, right FG: 40–52 −18). We then placed spheres with a radius of 12 mm around these four coordinates to search for the participant-specific local maxima within each participant's session-specific SPM for the faces minus shapes contrast (adjusted for the $t$ effects of interest, $p < 0.05$ uncorrected). Note that for the purposes of extracting the fMRI data for the DCM nodes, one does not need corrected p-values (*Hillebrandt et al., 2014*). Next, we defined the ROIs by a 6 mm radius sphere around the participant- and session-specific local maxima. For the subcortical areas of interest, we defined the initial search radius by the anatomically defined ROI masks (as described above) instead of significant peaks from the group analysis to confine our search within subcortical grey matter.

We used a 'two-state' DCM model, which accounts for both excitatory and inhibitory neural populations (*Hillebrandt et al., 2014*; *Marreiros et al., 2008*). Our model space was dictated by our specific, theory-driven hypotheses about subcortical and cortical visual pathways to the amygdala, as well as by the significant regions of the BOLD signal observed at the group level in our GLM analysis. Both face and shape blocks contributed to input parameters within each model. All endogenous and intrinsic connections in each model were modulated by the effect of faces over shapes.

To specify a DCM, each participant needed to have above-threshold activation (at $p < 0.05$, uncorrected) within each ROI across both scanning sessions. This was the case for 237 out of the 622 participants (see *Appendix 1—figure 1*). The ROIs with the highest numbers of below-threshold participants were the left and right superior colliculi (261 and 246 participants, respectively), followed by the left and right pulvinar (46 and 32 participants, respectively), and finally the left and right amygdala (40 and 25 participants, respectively). This may be due to the bilateral superior colliculi's relatively smaller volume as well as lower statistical power (its mean $t$-statistic was approximately 10.57 compared with 33.35 for IOG and 30.14 for FG). Critically, the group of 237 participants with above-threshold BOLD responses in all ROIs did not differ significantly from the other group of 385 participants in the global or local tractography results (main effect of 'group' and interactions with 'group' were all $p > 0.162$ and $\eta_p^2 <.003$), performance on the Penn Emotion Recognition task (all independent-samples $t$-tests had $p > 0.100$), volume of the thalamus (left: $p = 0.055$, right: $p = 0.987$)/amygdala (left: $p = 0.472$, right: $p = 0.394$)/fusiform area (left: $p = 0.677$, right: $p = 0.597$)/lateral occipital area (left: $p = 0.762$, right: $p = 0.679$; volumes computed by Freesurfer), median reaction time (faces: $p = 0.418$, shapes: $p = 0.617$) and accuracy (faces: $p = 0.417$, shapes: $p = 0.717$) during the fMRI task, age ($p = 0.782$), or gender ($p = 0.359$). Therefore, using the information available to us, we had no evidence to assume that our DCM sample was biased by any confounding variable.

The final model space consisted of 102 models (see *Figure 5*), where the first Cortical family contained six models, the second and third Cortical families contained 12 models each, and the Dual families (families 4, 5, and 6) contained 24 models each. The different families correspond to different input types (superior colliculus only, pulvinar only, or superior colliculus and pulvinar) and the different models within these families arise from different combinations of forward and backward connections. Each of the final 102 DCMs were modelled separately for both fMRI sessions. Both hemispheres were included in each model with no cross-hemispheric connections. To determine which model best explained the data, we conducted family-wise Bayesian Model Selection (*Stephan et al., 2009*; *Rigoux et al., 2014*), which penalises models for complexity according to the free energy principle (*Friston et al., 2006*). We used the random effects implementation to account for potential individual differences in the recruitment of a subcortical pathway for viewing faces (*Stephan et al., 2009*).

## Code availability

All computer codes that were used to produce the results (from raw HCP data to track counts, fibre density, BOLD signal and DCM files) is freely available online via GitHub (*McFadyen, 2018*; copy archived at https://github.com/elifesciences-publications/hcp-diffusion-dcm) and the Open Science Framework (doi:10.17605/OSF.IO/KBPWM).

## Data availability

The data analysed in this study came from the publicly-available Human Connectome Project S900 release: https://www.humanconnectome.org/study/hcp-young-adult/document/900-subjects-data-release. Restricted access was obtained through the HCP to acquire specific participant ages (in years) and drug/alcohol information. Ethical permission was granted by the University of Queensland Human Research Ethics Committee. No figures display raw data.

## Acknowledgements

Data were provided by the Human Connectome Project, WU-Minn Consortium (Principal Investigators: David Van Essen and Kamil Ugurbil; 1U54MH091657) funded by the 16 NIH Institutes and Centers that support the NIH Blueprint for Neuroscience Research; and by the McDonnell Center for Systems Neuroscience at Washington University. This work was funded by a University of Queensland Fellowship (2016000071) to MIG, an Australian Research Council (ARC) Australian Laureate Fellowship (FL110100103) and an ARC Centre of Excellence for Integrative Brain Function grant (CE140100007) to MIG and JBM, and an ARC Australian Postgraduate Award to JM.

## Additional information

### Funding

| Funder | Grant reference number | Author |
|---|---|---|
| Australian Research Council | Australian Postgraduate Award | Jessica McFadyen |
| Australian Research Council | Top-Up Scholarship | Jessica McFadyen |
| Australian Research Council | CE140100007 | Jason B Mattingley Marta I Garrido |
| Australian Research Council | Australian Laureate Fellowship (FL110100103) | Jason B Mattingley |
| University of Queensland | UQ Fellowship,2016000071 | Marta I Garrido |

The funders had no role in study design, data collection and interpretation, or the decision to submit the work for publication.

### Author contributions

Jessica McFadyen, Data curation, Software, Formal analysis, Validation, Visualization, Methodology, Writing—original draft; Jason B Mattingley, Supervision, Project administration, Writing—review and editing; Marta I Garrido, Conceptualization, Methodology, Supervision, Funding acquisition, Project administration, Writing—review and editing

### Author ORCIDs

Jessica McFadyen  http://orcid.org/0000-0003-1415-2286
Jason B Mattingley  http://orcid.org/0000-0003-0929-9216
Marta I Garrido  https://orcid.org/0000-0003-0679-4959

### Ethics

Human subjects: Ethics for data acquisition were obtained by the Washington University - University of Minnesota Consortium of the Human Connectome Project (WU-Minn HCP). All participants gave written consent to participate in the study. The University of Queensland Human Research Ethics Committee (2014001212) approved the Open and Restricted Access to the HCP dataset.

### Decision letter and Author response

Decision letter https://doi.org/10.7554/eLife.40766.027
Author response https://doi.org/10.7554/eLife.40766.028

# Additional files

## Supplementary files

• Transparent reporting form
DOI: https://doi.org/10.7554/eLife.40766.009

## Data availability

The data analysed in this study came from the publicly-available Human Connectome Project S900 release: https://www.humanconnectome.org/study/hcp-young-adult/document/900-subjects-data-release. Restricted access was obtained through the HCP to acquire specific participant ages (in years) and drug/alcohol information. Computer code used to produce the results is available on GitHub (https://github.com/jjmcfadyen/hcp-diffusion-dcm; copy archived at https://github.com/eli-fesciences-publications/hcp-diffusion-dcm).

The following datasets were generated:

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

# Appendix 1

DOI: https://doi.org/10.7554/eLife.40766.010

## Supplementary materials

### Dynamic causal modelling

Within the DCM subsample of 237 participants, 49 were unrelated. For this sample, the winning family was the "Dual with SC and PUL input" (expected probability = 64.86%, exceedance probability = 99.92%). The winning model was within this family (expected probability = 21.15%, exceedance probability = 72.77%) and was the same as the winning model from the full 237 participant sample. Classical statistics on the exponentiated parameter estimates showed that all B parameter estimates were significant except the backward connections between right and left FG to IOG, as was found in the full sample.

We conducted the same series of eight correlations between the structural connectivity measures (global track count and summed weights) and effective connectivity (A and B parameter estimates), while also removing 1 outlier. After correcting for multiple comparisons, we found that participants with greater global fibre count along the right PUL-AMG connection also had stronger modulatory connectivity ($r = .422$, $p_{bonf} = .025$) along the same connection.

### Comparison of tractographically-reconstructed fibres

#### SC-PUL and PUL-AMG

Repeated measures ANOVAs were conducted to compare the fibre counts (global tractography) or apparent fibre density (local tractography) between pathways (SC-PUL, PUL-AMG) and hemispheres (left, right). Absolute relative head motion was included as a covariate of no interest in each test. Greenhouse Geisser corrections made when Mauchly's test of sphericity was significant. All confidence intervals adjusted for Bonferroni correction. Follow-up paired t-tests conducted for significant interactions, bootstrapped with 1,000 iterations. Outliers were removed according to whether participants had data on at least one variable with a standardised residual score above or below 3.

**Appendix 1—table 1.** Global tractography fibre counts.
2 (pathway: SC-PUL, PUL-AMG)×2 (hemisphere: left, right) design. Four outliers removed out of 622.

| Outliers removed | ANOVA main effects | Effect | Df | F | P | $\eta_p^2$ | t-tests for Simple Effects | | |
|---|---|---|---|---|---|---|---|---|---|
| | | | | | | | Df | T | P |
| | | * Pathway | (3,4) | 433.286 | $2.842 \times 10^{-73}$* | 0.413 | | | |
| | | * Hemisphere | (3,4) | 7.583 | 0.006 * | 0.012 | | | |
| | | * Pathway × Hemisphere | (1,616) | 16.025 | $7.000 \times 10^{-5}$* | 0.025 | | | |
| | | Left vs.Right SC-PUL | | | | | 617 | 1.013 | .311 |
| | | * Left vs. Right PUL-AMG | | | | | 617 | −9.785 | $4.070 \times 10^{-21}$* |
| | | Pathway × Head Motion | (1,616) | 0.313 | .576 | $5.090 \times 10^{-4}$ * | | | |
| | | Hemisphere × Head Motion | (1,616) | 0.055 | .813 | $9.100 \times 10^{-5}$ * | | | |
| | | Pathway × Hemisphere × Head Motion | (1,616) | 0.332 | .710 | $2.250 \times 10^{-4}$ * | | | |
| No Outliers Removed | | * Pathway | (1,620) | 32.416 | $2.194 \times 10^{-8}$* | .049 | | | |
| | | * Hemisphere | (1,620) | 9.175 | .003 * | .015 | | | |
| | | * Pathway × Hemisphere | (1,620) | 5.807 | .016 * | .009 | | | |
| | | Left vs. Right SC-PUL | | | | | 621 | 0.654 | .514 |
| | | * Left vs. Right PUL-AMG | | | | | 621 | −8.904 | $5.906 \times 10^{-18}$* |
| | | Pathway × Head Motion | (1,620) | 0.030 | .863 | $4.800 \times 10^{-5}$ | | | |
| | | Hemisphere × Head Motion | (1,620) | 1.155 | .283 | .002 | | | |
| | | Pathway × Hemisphere × Head Motion | (1,620) | 0.332 | .565 | .001 | | | |

*$p$ <.05

DOI: https://doi.org/10.7554/eLife.40766.011

**Appendix 1—table 2.** Local tractography fibre counts.
2 (pathway: SC-PUL, PUL-AMG)×2 (hemisphere: left, right) design. Thirteen outliers removed out of 622.

| | ANOVA main effects | | | | | t-tests for Simple Effects | | |
| | Effect | Df | F | P | $\eta_p^2$ | Df | T | P |
|---|---|---|---|---|---|---|---|---|
| Outliers removed | * Pathway | (1,607) | 69.586 | $4.930 \times 10^{-16}$* | 0.103 | | | |
| | Hemisphere | (1,607) | 1.387 | 0.239 | 0.002 | | | |
| | * Pathway × Hemisphere | (1,607) | 162.48 | $3.828 \times 10^{-33}$* | 0.211 | | | |
| | * Left vs. Right SC-PUL | | | | | 608 | 10.749 | $8.596 \times 10^{-25}$* |
| | * Left vs. Right PUL-AMG | | | | | 608 | −18.205 | $1.960 \times 10^{-59}$* |
| | Pathway × Head Motion | (1,607) | 1.815 | .178 | .003 | | | |
| | Hemisphere × Head Motion | (1,607) | 1.39 | .239 | .002 | | | |
| | Pathway × Hemisphere × Head Motion | (1,607) | 1.425 | .233 | .002 | | | |
| No Outliers Removed | * Pathway | (1,620) | 59.749 | $4.365 \times 10^{-14}$* | .088 | | | |
| | Hemisphere | (1,620) | 0.757 | .385 | .001 | | | |
| | * Pathway ×Hemisphere | (1,620) | 116.962 | $4.316 \times 10^{-25}$* | .159 | | | |
| | * Left vs. Right SC-PUL | | | | | 621 | 9.378 | $1.243 \times 10^{-19}$* |
| | * Left vs. Right PUL-AMG | | | | | 621 | −17.515 | $4.068 \times 10^{-56}$* |
| | Pathway × Head Motion | (1,620) | 1.222 | .269 | .002 | | | |
| | Hemisphere × Head Motion | (1,620) | 1.511 | .219 | .002 | | | |
| | Pathway × Hemisphere × Head Motion | (1,620) | 0.617 | .433 | .001 | | | |

*$p <.05$

DOI: https://doi.org/10.7554/eLife.40766.012

**Appendix 1—table 3.** Global tractography fibre counts (terminations in pulvinar subregions) for SC-PUL. 5 (cluster: inferior, medial, anterior, superior, lateral)×2 (hemisphere: left, right) design. Ten outliers removed out of 622.

Outliers removed

| | | ANOVA main effects | | | | t-tests for Simple Effects | | |
| --- | --- | --- | --- | --- | --- | --- | --- | --- |
| | Effect | Df | F | P | $\eta_p^2$ | Df | T | P |
| | Hemisphere | (1,610) | 3.817 | .051 | .090 | | | |
| | * Cluster | (3,2060) | 214.015 | $1.584 \times 10^{-133}$* | .260 | | | |
| | *Hemisphere × Cluster | (3,2014) | 23.624 | $2.870 \times 10^{-16}$* | .037 | | | |
| | * Left: Inferior vs. Anterior | | | | | 611 | 4.820 | $1.815 \times 10^{-6}$* |
| | * Left: Anterior vs. Medial | | | | | 611 | 2.754 | 0.006 * |
| | * Left: Medial vs. Lateral | | | | | 611 | 11.635 | $2.114 \times 10^{-28}$* |
| | * Left: Lateral vs. Superior | | | | | 611 | 15.938 | $4.456 \times 10^{-48}$* |
| | * Right:Anterior vs. Inferior | | | | | 611 | 4.228 | $2.715 \times 10^{-5}$* |
| | * Right: Inferior vs. Lateral | | | | | 611 | 13.982 | $9.574 \times 10^{-39}$* |
| | Right: Lateral vs. Medial | | | | | 611 | 2.045 | 0.041 |
| | * Right: Medial vs. Superior | | | | | 611 | 21.108 | $1.076 \times 10^{-74}$* |
| | Hemisphere × Head Motion | (1,610) | 0.620 | 0.431 | .001 | | | |
| | Cluster × Head Motion | (3,2060) | 0.478 | 0.720 | .001 | | | |
| | Hemisphere × Cluster × Head Motion | (3,2014) | 0.306 | 0.840 | .001 | | | |

*Appendix 1—table 3 continued on next page*

| No Outliers Removed | df | $F$ / $t$ | $p$ | | $p$ (t-tests) |
|---|---|---|---|---|---|
| * Hemisphere | (1,620) | 5.749 | .017 * | .009 | |
| * Cluster | (1,913) | 52.174 | $2.998 \times 10^{-17}$* | .078 | |
| *Hemisphere × Cluster | (3,1743) | 16.591 | $4.129 \times 10^{-10}$* | .026 | |
| * Left: Inferior vs. Anterior | 621 | 4.191 | | | $3.200 \times 10^{-5}$ * |
| Left: Anterior vs. Medial | 621 | 2.314 | | | .021 |
| * Left: Medial vs. Lateral | 621 | 6.385 | | | $3.361 \times 10^{-10}$ * |
| * Left: Lateral vs. Superior | 621 | 14.235 | | | $5.357 \times 10^{-40}$ * |
| * Right: Anterior vs. Inferior | 621 | 3.695 | | | $2.390 \times 10^{-5}$ * |
| * Right: Inferior vs. Lateral | 621 | 7.619 | | | $9.590 \times 10^{-14}$ * |
| Right:Lateral vs. Medial | 621 | −0.187 | | | .852 |
| * Right: Medial vs. Superior | 621 | 12.957 | | | $3.715 \times 10^{-34}$ * |
| Hemisphere × Head Motion | (1,620) | 1.555 | .213 | .003 | |
| Cluster × Head Motion | (1,913) | 0.159 | .786 | $2.570 \times 10^{-4}$ | |
| Hemisphere × Cluster × Head Motion | (3,1743) | 0.706 | .539 | .001 | |

*$p$ <.05 (for t-tests, only * if <.05 Bonferroni corrected)

DOI: https://doi.org/10.7554/eLife.40766.013

**Appendix 1—table 4.** Global tractography fibre counts (terminations in pulvinar subregions) for PUL-AMG. 5 (cluster: inferior, medial, anterior, superior, lateral)×2 (hemisphere: left, right) design. Sixty-eight outliers removed out of 622.

Outliers removed

| ANOVA main effects | | | | | t-tests for Simple Effects | | |
| --- | --- | --- | --- | --- | --- | --- | --- |
| Effect | Df | F | P | $\eta_p^2$ | Df | T | P |
| * Hemisphere | (1,552) | 22.147 | $3.197 \times 10^{-6}$* | .039 | | | |
| * Cluster | (2,1139) | 354.261 | $2.662 \times 10^{-123}$* | .391 | | | |
| *Hemisphere × Cluster | (2,1135) | 15.334 | $1.946 \times 10^{-7}$* | .027 | | | |
| * Left: Inferior vs. Medial | | | | | 533 | 24.849 | $4.248 \times 10^{-92}$* |
| * Left: Medial vs. Lateral | | | | | 533 | 4.209 | $2.998 \times 10^{-5}$* |
| Left: Lateral vs. Anterior | | | | | 533 | 2.458 | .014 |
| * Left: Anterior vs. Superior | | | | | 533 | 4.665 | $3.868 \times 10^{-6}$* |
| * Right:Inferior vs. Lateral | | | | | 533 | 25.742 | $1.184 \times 10^{-96}$* |
| * Right: Lateral vs. Superior | | | | | 533 | 7.609 | $1.197 \times 10^{-13}$* |
| Right: Superior vs. Medial | | | | | 533 | 2.044 | .041 |
| Right: Medial vs. Anterior | | | | | 533 | 4.133 | $4.141 \times 10^{-5}$* |
| Hemisphere × Head Motion | (1,552) | 1.067 | .302 | .002 | | | |
| Cluster × Head Motion | (2,1139) | 1.885 | .151 | .003 | | | |
| Hemisphere ×Cluster × Head Motion | (2,1135) | 0.601 | .553 | .001 | | | |

| No Outliers Removed | | df | F/t | p | |
|---|---|---|---|---|---|
| | * Hemisphere | (1,620) | 33.464 | $1.152 \times 10^{-8}$ * | .051 |
| | * Cluster | (2,1153) | 320.406 | $3.145 \times 10^{-105}$ * | .341 |
| | *Hemisphere × Cluster | (2,1390) | 14.935 | $9.810 \times 10^{-8}$ * | .024 |
| | * Left:Inferior vs. Anterior | 621 | 23.865 | $8.002 \times 10^{-90}$ * | |
| | Left: Anterior vs. Medial | 621 | 3.099 | .002 | |
| | * Left: Medial vs. Lateral | 621 | 3.021 | .003 | |
| | * Left:Lateral vs. Superior | 621 | 3.835 | $1.380 \times 10^{-4}$ * | |
| | * Right: Anterior vs. Inferior | 621 | 25.424 | $2.827 \times 10^{-98}$ * | |
| | * Right: Inferior vs. Lateral | 621 | 7.909 | $1.190 \times 10^{-14}$ * | |
| | Right: Lateral vs. Medial | 621 | 0.657 | .511 | |
| | * Right: Medial vs. Superior | 621 | 3.925 | $9.600 \times 10^{-5}$ * | |
| | Hemisphere × Head Motion | (1,620) | 0.278 | .598 | $4.490 \times 10^{-4}$ |
| | Cluster × Head Motion | (2,1153) | 1.134 | .319 | .002 |
| | Hemisphere × Cluster × Head Motion | (2,1390) | 0.512 | .620 | .001 |

*p <.05 (for t-tests, only * if <0.05 Bonferroni corrected).

DOI: https://doi.org/10.7554/eLife.40766.014

**Appendix 1—table 5.** Global tractography fibre counts (terminations in amygdala subregions) for PUL-AMG. 3 (cluster: centromedial, basolateral, superficial)×2 (hemisphere: left, right) design. Sixteen outliers removed.

Outliers removed

| ANOVA main effects | | | | | t-tests for Simple Effects | | |
|---|---|---|---|---|---|---|---|
| Effect | Df | F | P | $\eta_p^2$ | Df | T | P |
| Hemisphere | (1,604) | 1.419 | .234 | .002 | | | |
| * Cluster | (2,1161) | 80.779 | $1.909 \times 10^{-32}$ * | .118 | | | |
| * Hemisphere × Cluster | (2,1171) | 28.278 | $2.060 \times 10^{-12}$ * | .045 | | | |
| * Left: Basolateral vs. Centromedial | | | | | 605 | 8.741 | $2.279 \times 10^{-17}$ * |
| * Left: Centromedial vs. Superficial | | | | | 605 | 5.354 | $1.222 \times 10^{-7}$ * |
| * Right: Basolateral vs. Superficial | | | | | 605 | 11.515 | $7.018 \times 10^{-28}$ * |
| * Right: Superficial vs. Centromedial | | | | | 605 | 10.700 | $1.376 \times 10^{-24}$ * |
| Hemisphere × Head Motion | (1,604) | 0.586 | .444 | $.695 \times 10^{-4}$ | | | |
| Cluster × Head Motion | (2,1161) | 0.892 | .407 | .001 | | | |
| Hemisphere × Cluster × Head Motion | (2,1171) | 0.261 | .764 | $4.315 \times 10^{-4}$ | | | |

*Appendix 1—table 5 continued on next page*

| No Outliers Removed | df | F / t | p | partial η² |
|---|---|---|---|---|
| Hemisphere | (1,620) | 2.295 | .130 | .004 |
| * Cluster | (2,1212) | 79.253 | $2.017 \times 10^{-32}$ * | .113 |
| * Hemisphere × Cluster | (2,1201) | 27.754 | $3.363 \times 10^{-12}$ * | .043 |
| * Left: Basolateral vs. Centromedial | 621 | 8.806 | $1.284 \times 10^{-17}$ * | |
| * Left: Centromedial vs. Superficial | 621 | 5.119 | $4.098 \times 10^{-7}$ * | |
| * Right: Basolateral vs. Superficial | 621 | 11.753 | $6.132 \times 10^{-29}$ * | |
| * Right: Superficial vs. Centromedial | 621 | 10.027 | $4.961 \times 10^{-22}$ * | |
| Hemisphere × Head Motion | (1,620) | 0.014 | .907 | $2.200 \times 10^{-4}$ |
| Cluster × Head Motion | (2,1212) | 0.515 | .593 | .001 |
| Hemisphere × Cluster × Head Motion | (2,1201) | 0.164 | .842 | $2.650 \times 10^{-4}$ |

*p <.05 (for t-tests, only * if <0.05 Bonferroni corrected)

DOI: https://doi.org/10.7554/eLife.40766.015

**Appendix 1—table 6.** Local tractography fibre counts (terminations in pulvinar subregions) for SC-PUL. 5 (cluster: inferior, medial, anterior, superior, lateral)×2 (hemisphere: left, right) design. Sixty-seven outliers removed out of 622.

| Effect | | ANOVA main effects | | | | t-tests for Simple Effects | | |
|---|---|---|---|---|---|---|---|---|
| | Df | F | P | $\eta_P^2$ | | Df | T | P |
| * Hemisphere | (1,553) | 23.577 | $1.564 \times 10^{-6}$ * | .041 | | | | |
| * Cluster | (2,1010) | 873.922 | $6.337 \times 10^{-209}$ * | .612 | | | | |
| *Hemisphere × Cluster | (2,1176) | 64.370 | $1.002 \times 10^{-28}$ * | .104 | | | | |
| * Left: Anterior vs. Inferior | | | | | | 554 | 22.373 | $1.709 \times 10^{-79}$ * |
| * Left: Inferior vs. Medial | | | | | | 554 | 9.742 | $8.353 \times 10^{-21}$ * |
| * Left:Medial vs. Lateral | | | | | | 554 | 30.710 | $1.089 \times 10^{-121}$ * |
| *Left:Lateral vs. Superior | | | | | | 554 | 16.514 | $4.103 \times 10^{-50}$ * |
| * Right:Anterior vs. Inferior | | | | | | 554 | 39.241 | $4.436 \times 10^{-162}$ * |
| * Right: Inferior vs. Medial | | | | | | 554 | 9.157 | $1.022 \times 10^{-18}$ * |
| * Right: Medial vs. Lateral | | | | | | 554 | 25.287 | $2.203 \times 10^{-94}$ * |
| * Right:Lateral vs. Superior | | | | | | 554 | 15.213 | $6.282 \times 10^{-44}$ * |
| Hemisphere × Head Motion | (1,553) | 0.358 | 0.550 | .001 | | | | |
| Cluster × Head Motion | (2,1010) | 0.495 | 0.593 | .001 | | | | |
| Hemisphere × Cluster × Head Motion | (2,1176) | 0.604 | 0.557 | .001 | | | | |

*Appendix 1—table 6 continued on next page*

| | | | | | |
|---|---|---|---|---|---|
| No Outliers Removed | * Hemisphere | (1,620) | 23.600 | $2.000 \times 10^{-6}$ * | .037 |
| | * Cluster | (2,1244) | 831.764 | $1.817 \times 10^{-230}$ * | .573 |
| | *Hemisphere × Cluster | (2,1354) | 69.262 | $1.107 \times 10^{-31}$ * | .100 |
| | * Left: Anterior vs. Inferior | 621 | 22.745 | $9.069 \times 10^{-84}$ * | |
| | * Left:Inferior vs. Medial | 621 | 8.436 | $2.312 \times 10^{-16}$ * | |
| | * Left:Medial vs. Lateral | 621 | 29.562 | $1.448 \times 10^{-120}$ * | |
| | *Left:Lateral vs. Superior | 621 | 13.832 | $3.972 \times 10^{-39}$ * | |
| | * Right: Anterior vs. Inferior | 621 | 39.100 | $1.326 \times 10^{-169}$ * | |
| | * Right:Inferior vs. Medial | 621 | 7.436 | $3.463 \times 10^{-13}$ * | |
| | * Right:Medial vs. Lateral | 621 | 23.544 | $4.380 \times 10^{-88}$ * | |
| | * Right:Lateral vs. Superior | 621 | 12.964 | $3.461 \times 10^{-34}$ * | |
| | Hemisphere × Head Motion | (1,620) | 0.161 | .688 | $2.60 \times 10^{-4}$ |
| | Cluster × Head Motion | (2,1244) | 0.830 | .437 | .001 |
| | Hemisphere × Cluster × Head Motion | (2,1354) | 1.023 | .365 | .002 |

*p <.05 (for t-tests, only * if <0.05 Bonferroni corrected)

DOI: https://doi.org/10.7554/eLife.40766.016

**Appendix 1—table 7.** Local tractography fibre counts (terminations in pulvinar subregions) for AMG-PUL. 5 (cluster: inferior, medial, anterior, superior, lateral)×2 (hemisphere: left, right) design. Seventy-eight outliers removed out of 622.

Outliers removed

| ANOVA main effects | | | | | t-tests for Simple Effects | | |
|---|---|---|---|---|---|---|---|
| Effect | Df | F | P | $\eta_p^2$ | Df | T | P |
| * Hemisphere | (1,542) | 108.301 | $3.019 \times 10^{-23}$ * | .167 | | | |
| * Cluster | (1,544) | 2159.827 | $9.415 \times 10^{-192}$ * | .799 | | | |
| *Hemisphere × Cluster | (1,544) | 105.772 | $7.093 \times 10^{-23}$ * | .163 | | | |
| * Left: Inferior vs. Lateral | | | | | 543 | 57.694 | $8.996 \times 10^{-234}$ * |
| * Left: Lateral vs. Medial | | | | | 543 | 10.633 | $4.080 \times 10^{-24}$ * |
| * Left: Medial vs. Anterior | | | | | 543 | 12.744 | $9.913 \times 10^{-33}$ * |
| * Left: Anterior vs. Superior | | | | | 543 | 5.897 | $6.503 \times 10^{-9}$ * |
| * Right: Inferior vs. Lateral | | | | | 543 | 59.079 | $1.332 \times 10^{-238}$ * |
| * Right: Lateral vs. Superior | | | | | 543 | 27.277 | $7.886 \times 10^{-104}$ * |
| * Right: Superior vs. Anterior | | | | | 543 | 14.006 | $2.839 \times 10^{-38}$ * |
| Right: Anterior vs. Medial | | | | | 543 | 0.815 | .416 |
| Hemisphere × Head Motion | (1,542) | 4.749 | .030 * | 0.009 | | | |
| Cluster × Head Motion | (1,544) | 19.569 | $1.146 \times 10^{-5}$ * | 0.035 | | | |
| Hemisphere × Cluster × Head Motion | (1,544) | 4.689 | .031 * | 0.009 | | | |

| | | df | F | p | η²p | df | t | p |
|---|---|---|---|---|---|---|---|---|
| No Outliers Removed | * Hemisphere | (1,620) | 110.587 | $6.523 \times 10^{-24}$ * | .151 | | | |
| | * Cluster | (1,622) | 2139.649 | $4.887 \times 10^{-204}$ * | .775 | | | |
| | *Hemisphere × Cluster | (1,624) | 107.010 | $2.219 \times 10^{-23}$ * | .147 | | | |
| | * Left: Inferior vs. Lateral | | | | | 621 | 57.120 | $2.174 \times 10^{-249}$ * |
| | * Left: Lateral vs. Medial | | | | | 621 | 10.169 | $1.427 \times 10^{-22}$ * |
| | * Left:Medial vs. Anterior | | | | | 621 | 10.609 | $2.812 \times 10^{-24}$ * |
| | *Left: Anterior vs. Superior | | | | | 621 | 5.153 | $3.446 \times 10^{-7}$ * |
| | * Right: Inferior vs. Lateral | | | | | 621 | 61.529 | $1.963 \times 10^{-266}$ * |
| | * Right:Lateral vs. Superior | | | | | 621 | 24.241 | $7.327 \times 10^{-92}$ * |
| | Right: Superior vs. Anterior | | | | | 621 | 10.891 | $2.122 \times 10^{-25}$ * |
| | Right: Anterior vs. Medial | | | | | 621 | 0.227 | .821 |
| | Hemisphere × Head Motion | (1,620) | 4.775 | .029 * | .147 | | | |
| | Cluster × Head Motion | (1,622) | 19.739 | $1.000 \times 10^{-5}$ * | .031 | | | |
| | Hemisphere × Cluster × Head Motion | (1,624) | 4.568 | .033 | .007 | | | |

*p <.05 (for t-tests, only * if <0.05 Bonferroni corrected)

DOI: https://doi.org/10.7554/eLife.40766.017

**Appendix 1—table 8.** Local tractography fibre counts (terminations in amygdala subregions) for PUL-AMG. 3 (cluster: centromedial, basolateral, superficial)×2 (hemisphere: left, right) design. Thirty-seven outliers removed out of 622.

Outliers removed

| Effect | ANOVA main effects | | | | t-tests for Simple Effects | | |
|---|---|---|---|---|---|---|---|
| | Df | F | P | $\eta_p^2$ | Df | T | P |
| Hemisphere | (1,620) | 511.278 | $5.154 \times 10^{-83}$ * | .452 | | | |
| * Cluster | (2,989) | 308.079 | $7.593 \times 10^{-88}$ * | .332 | | | |
| *Hemisphere × Cluster | (2,1201) | 27.754 | $3.363 \times 10^{-12}$ * | .043 | | | |
| * Left: Basolateral vs. Centremedial | | | | | 621 | 10.374 | $2.323 \times 10^{-23}$ * |
| * Left: Centromedial vs. Superficial | | | | | 621 | 19.573 | $8.248 \times 10^{-67}$ * |
| * Right: Centromedial vs. Basolateral | | | | | 621 | 25.136 | $1.037 \times 10^{-96}$ * |
| * Right: Basolateral vs. Superficial | | | | | 621 | 27.037 | $5.308 \times 10^{-107}$ * |
| Hemisphere × Head Motion | (1,620) | 296.624 | .015 * | .009 | | | |
| Cluster × Head Motion | (2,989) | 0.050 | $4.774 \times 10^{-84}$ | .324 | | | |
| Hemisphere × Cluster × Head Motion | (2,1201) | 0.917 | .380 | .001 | | | |

| No Outliers Removed | df | | | | df | | |
|---|---|---|---|---|---|---|---|
| Hemisphere | (1,620) | 511.278 | $5.154 \times 10^{-83}$ * | .452 | | | |
| * Cluster | (2,989) | 308.079 | $7.593 \times 10^{-88}$ * | .332 | | | |
| *Hemisphere × Cluster | (2,1201) | 27.754 | $3.363 \times 10^{-12}$ * | .043 | | | |
| * Left: Basolateral vs. Centremedial | | | | | 621 | 10.374 | $2.323 \times 10^{-23}$ * |
| * Left: Centromedial vs. Superficial | | | | | 621 | 19.573 | $8.248 \times 10^{-67}$ * |
| * Right: Centromedial vs. Basolateral | | | | | 621 | 25.136 | $1.037 \times 10^{-96}$ * |
| * Right: Basolateral vs. Superficial | | | | | 621 | 27.037 | $5.308 \times 10^{-107}$ * |
| Hemisphere × Head Motion | (1,620) | 296.624 | .015 * | .009 | | | |
| Cluster × Head Motion | (2,989) | 0.050 | $4.774 \times 10^{-84}$ | .324 | | | |
| Hemisphere × Cluster × Head Motion | (2,1201) | 0.917 | .380 | .001 | | | |

*p <.05 (for t-tests, only * if <0.05 Bonferroni corrected)

DOI: https://doi.org/10.7554/eLife.40766.018

**Appendix 1—table 9.** Paired *t*-tests between local streamline count and null distribution.

| Comparison | N | M | SEM | 95% | T | Df | P |
|---|---|---|---|---|---|---|---|
| Left SC-PUL | 618 | 1384.876 | 16.704 | [1352.073, 1417.68] | 82.907 | 617 | $<1 \times 10^{-243}$ * |
| Right SC-PUL | 618 | −499.555 | 16.430 | [−531.821, −467.289] | −30.404 | 617 | $8.832 \times 10^{-125}$ * |
| Left PUL-AMG | 618 | 351.669 | 8.535 | [334.908, 368.431] | 41.202 | 617 | $2.722 \times 10^{-179}$ * |
| Right PUL-AMG | 620 | 577.535 | 10.385 | [557.142, 597.929] | 55.614 | 619 | $6.091 \times 10^{-243}$ * |

*$p <.05$

DOI: https://doi.org/10.7554/eLife.40766.019

**Appendix 1—table 10.** Global tractography multivariate GLM.
Fibre counts for left and right SC-PUL and PUL-AMG entered as dependent variables. Recognition scores (out of eight) for fearful, sad, and angry expressions entered as covariates of interest. Head motion entered as covariate of no interest. Bootstrapping at 1000 iterations. Four outliers removed with residuals more than three standard deviations from the mean.

Outliers removed

| Multivariate tests | | | | | |
|---|---|---|---|---|---|
| Effect | Wilk's Λ | Df | F | P | $\eta_p^2$ |
| Fearful | .995 | (4, 609) | 0.715 | .582 | .005 |
| Sad | .994 | (4, 609) | 0.915 | .455 | .006 |
| Angry | .998 | (4, 609) | 0.294 | .882 | .002 |
| Head Motion | .999 | (4, 609) | 0.223 | .926 | .001 |

| Parameter estimates | | | | | | |
|---|---|---|---|---|---|---|
| Dependent variable | Parameter | B | T | P | $\eta_p^2$ | 95% |
| Left SC-PUL | Fearful | −0.350 | −1.583 | .114 | .004 | [−0.784, 0.084] |
| | Sad | −0.312 | −1.520 | .129 | .004 | [−0.715, 0.091] |
| | Angry | −0.157 | −0.687 | .493 | .001 | [−0.607, 0.292] |
| | Head Motion | −0.359 | −0.677 | .498 | .001 | [−1.401, 0.682] |

| | Parameter | B | t | p | $\eta_p^2$ | 95% CI |
|---|---|---|---|---|---|---|
| Right SC-PUL | Fearful | −0.208 | −0.933 | .351 | .001 | [−0.645, 0.23] |
| | Sad | −0.042 | −0.202 | .840 | $6.700 \times 10^{-5}$ | [−0.448, 0.364] |
| | Angry | −0.046 | −0.197 | .844 | $6.400 \times 10^{-5}$ | [−0.499, 0.408] |
| | Head Motion | −0.268 | −0.501 | .616 | $4.110 \times 10^{-4}$ | [−1.317, 0.781] |
| Left PUL-AMG | Fearful | −0.006 | −0.051 | .959 | $4.000 \times 10^{-6}$ | [−0.225, 0.213] |
| | Sad | −0.010 | −0.099 | .921 | $1.600 \times 10^{-5}$ | [−0.213, 0.193] |
| | Angry | −0.024 | −0.205 | .837 | $6.900 \times 10^{-5}$ | [−0.251, 0.203] |
| | Head Motion | 0.103 | 0.386 | .700 | $2.430 \times 10^{-4}$ | [−0.422, 0.628] |
| Left PUL-AMG | Fearful | −0.071 | −0.611 | .541 | .001 | [−0.297, 0.156] |
| | Sad | 0.101 | 0.941 | .347 | .001 | [−0.11, 0.311] |
| | Angry | −0.101 | −0.846 | .398 | .001 | [−0.336, 0.134] |
| | Head Motion | −0.103 | −0.373 | .709 | $2.280 \times 10^{-4}$ | [−0.647, 0.44] |

**No Outliers Removed**

Multivariate Tests

| Effect | Wilk's Λ | df | F | p | $\eta_p^2$ |
|---|---|---|---|---|---|
| Fearful | .998 | (4,613) | 0.349 | .845 | .002 |
| Sad | .994 | (4,613) | 0.957 | .430 | .006 |
| Angry | .996 | (4,613) | 0.569 | .685 | .004 |
| Head Motion | .998 | (4,613) | 0.287 | .886 | .002 |

Parameter Estimates

| Dependent Variable | Parameter | B | t | p | $\eta_p^2$ | 95% CI |
|---|---|---|---|---|---|---|
| Left SC-PUL | Fearful | −0.794 | −0.809 | .419 | .001 | [−2.723, 1.134] |
| | Sad | 0.053 | 0.058 | .954 | $5.000 \times 10^{-7}$ | [−1.736, 1.842] |
| | Angry | 0.057 | 0.056 | .955 | $5.000 \times 10^{-7}$ | [−1.938, 2.052] |
| | Head Motion | −0.147 | −0.062 | .950 | $6.000 \times 10^{-7}$ | [−4.763, 4.47] |

*Appendix 1—table 10 continued on next page*

| Region | Condition | | | | |
|---|---|---|---|---|---|
| Right SC-PUL | Fearful | −0.612 | −0.605 | .546 | .001 | [−2.597, 1.374] |
| | Sad | 0.482 | 0.514 | .607 | $4.290 \times 10^{-4}$ | [−1.36, 2.325] |
| | Angry | 0.314 | 0.3 | .764 | $1.460 \times 10^{-4}$ | [−1.74, 2.369] |
| | Head Motion | −0.703 | −0.29 | .772 | $1.370 \times 10^{-4}$ | [−5.458, 4.052] |
| Left PUL-AMG | Fearful | −0.072 | −0.433 | .665 | $3.050 \times 10^{-4}$ | [−0.4, 0.255] |
| | Sad | 0.018 | 0.117 | .907 | $2.200 \times 10^{-5}$ | [−0.286, 0.322] |
| | Angry | −0.109 | −0.631 | .529 | .001 | [−0.448, 0.23] |
| | Head Motion | 0.191 | 0.479 | .632 | $3.720 \times 10^{-4}$ | [−0.593, 0.976] |
| Left PUL-AMG | Fearful | −0.126 | −0.871 | .384 | .001 | [−0.41, 0.158] |
| | Sad | 0.12 | 0.894 | .372 | .001 | [−0.144, 0.384] |
| | Angry | −0.135 | −0.905 | .366 | .001 | [−0.429, 0.158] |
| | Head Motion | −0.012 | −0.033 | .973 | $2.000 \times 10^{-6}$ | [−0.692, 0.669] |

DOI: https://doi.org/10.7554/eLife.40766.020

**Appendix 1—table 11.** Local tractography multivariate GLM.

Average apparent fibre density for left and right SC-PUL and PUL-AMG entered as dependent variables. Recognition scores (out of eight) for fearful, sad, and angry expressions entered as covariates of interest. Head motion entered as covariate of no interest. Bootstrapping at 1000 iterations. Fifteen outliers removed with residuals more than three standard deviations from the mean.

| Outliers removed | | | | | | |
|---|---|---|---|---|---|---|
| Multivariate tests | | | | | | |
| Effect | | Wilk's Λ | Df | F | P | $\eta_p^2$ |
| * Fearful | | .984 | (4,598) | 2.501 | .042 * | .016 |
| Sad | | .991 | (4,598) | 1.363 | .245 | .009 |
| Angry | | .993 | (4,598) | 1.111 | .350 | .007 |
| * Head Motion | | .977 | (4,598) | 3.451 | .008 * | .023 |
| Parameter estimates | | | | | | |
| Dependent variable | Parameter | B | T | | P | $\eta_p^2$ |
| Left SC-PUL | Fearful | 0 | −0.003 | | 0.998 | $1.093 \times 10^{-8}$ |
| | Sad | 0.084 | 1.099 | | 0.272 | 0.002 |
| | Angry | −0.078 | −0.901 | | 0.368 | 0.001 |
| | Head Motion | −0.135 | −0.681 | | 0.496 | 0.001 |

| | 95% |
|---|---|
| | [−0.163, 0.162] |
| | [−0.066, 0.235] |
| | [−0.249, 0.093] |
| | [−0.526, 0.255] |

| | Parameter | B | t | p | 95% CI |
|---|---|---|---|---|---|
| Right SC-PUL | Fearful | −0.046 | −0.645 | 0.519 | [−0.185, 0.094] |
| | Sad | −0.008 | −0.123 | 0.902 | [−0.137, 0.121] |
| | Angry | 0.043 | 0.574 | 0.566 | [−0.104, 0.189] |
| | Head Motion | −0.146 | −0.859 | 0.39 | [−0.48, 0.188] |
| Left PUL-AMG | *fearful | 0.14 | 2.887 | 0.004 * | [0.045, 0.235] |
| | Sad | −0.084 | −1.855 | 0.064 | [−0.172, 0.005] |
| | Angry | −0.052 | −1.018 | 0.309 | [−0.152, 0.048] |
| | Head Motion | −0.21 | −1.802 | 0.072 | [−0.438, 0.019] |
| Right PUL AMG | *fearful | 0.143 | 2.523 | 0.012 * | [0.032, 0.255] |
| | Sad | −0.042 | −0.787 | 0.431 | [−0.145, 0.062] |
| | Angry | 0.041 | 0.686 | 0.493 | [−0.076, 0.158] |
| | * Head Motion | −0.484 | −3.55 | $4.150 \times 10^{-4}$ * | [−0.752, −0.216] |

**No Outliers Removed**

**Multivariate Tests**

| Effect | Wilk's Λ | df | F | p | $\eta_p^2$ |
|---|---|---|---|---|---|
| * Fearful | .983 | (4,613) | 2.669 | .031 * | .017 |
| Sad | .993 | (4,613) | 1.080 | .366 | .007 |
| Angry | .997 | (4,613) | 0.499 | .736 | .003 |
| * Head Motion | .982 | (4,613) | 2.792 | .026 * | .018 |

**Parameter Estimates**

| Dependent Variable | Parameter | B | t | p | $\eta_p^2$ | 95% CI |
|---|---|---|---|---|---|---|
| Left SC-PUL | Fearful | −0.006 | −0.051 | .960 | $4.000 \times 10^{-6}$ | [−0.231, 0.22] |
| | Sad | 0.121 | 1.138 | .255 | .002 | [−0.088, 0.33] |
| | Angry | −0.080 | −0.674 | .500 | .001 | [−0.313, 0.153] |
| | Head Motion | −0.063 | −0.229 | .819 | $8.500 \times 10^{-5}$ | [−0.602, 0.477] |

Appendix 1—table 11 continued on next page

| Region | Effect | | | | | |
|--------|--------|--------|--------|--------|-------------------|------------------|
| Right SC-PUL | Fearful | −0.097 | −1.037 | .300 | .002 | [−0.28, 0.086] |
| | Sad | 0.017 | 0.197 | .844 | $6.300 \times 10^{-5}$ | [−0.153, 0.187] |
| | Angry | 0.023 | 0.233 | .816 | $8.800 \times 10^{-5}$ | [−0.167, 0.212] |
| | Head Motion | −0.165 | −0.738 | .461 | .001 | [−0.604, 0.274] |
| Left PUL-AMG | * Fearful | 0.137 | 2.650 | .008 * | .011 | [0.036, 0.239] |
| | Sad | −0.075 | −1.563 | .119 | .004 | [−0.17, 0.019] |
| | Angry | −0.049 | −0.904 | .366 | .001 | [−0.154, 0.057] |
| | Head Motion | −0.193 | −1.554 | .121 | .004 | [−0.437, 0.051] |
| Left PUL-AMG | * Fearful | 0.137 | 2.334 | .020 * | .009 | [0.022, 0.253] |
| | Sad | −0.053 | −0.976 | .330 | .002 | [−0.16, 0.054] |
| | Angry | 0.001 | 0.019 | .985 | $6.126 \times 10^{-7}$ | [−0.118, 0.121] |
| | * Head Motion | −0.465 | −3.303 | .001 * | .017 | [−0.741, −0.189] |

DOI: https://doi.org/10.7554/eLife.40766.021

**Appendix 1—table 12.** Partial correlations between fibre density of each pathway and the corresponding DCM parameter estimate. Scatterplots show the fibre density (x-axis) residuals (after regressing against head motion) and the DCM parameter estimate (y-axis) residuals (after regressing against head motion). *p<0.05, Bonferroni-corrected

Global Tractography

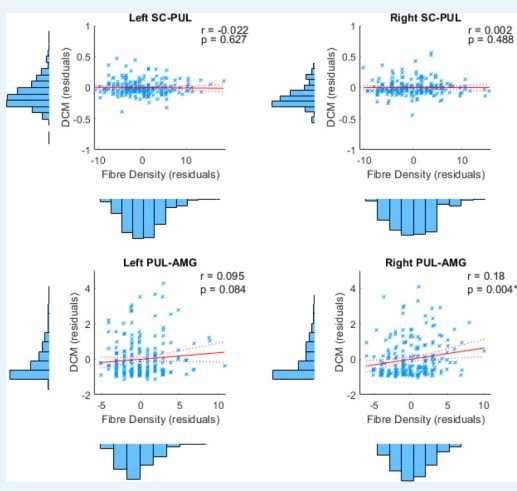

Local Tractography

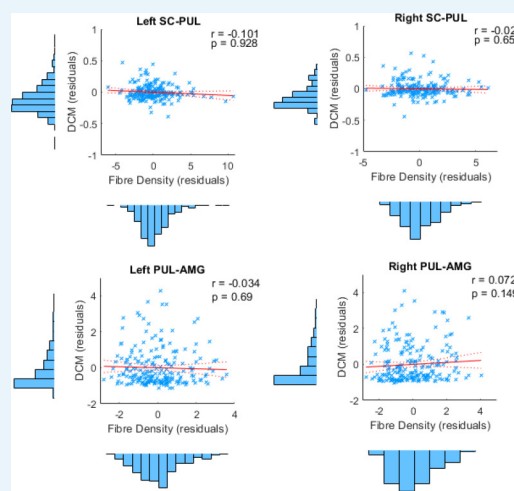

Global Tractography

*Appendix 1—table 12 continued on next page*

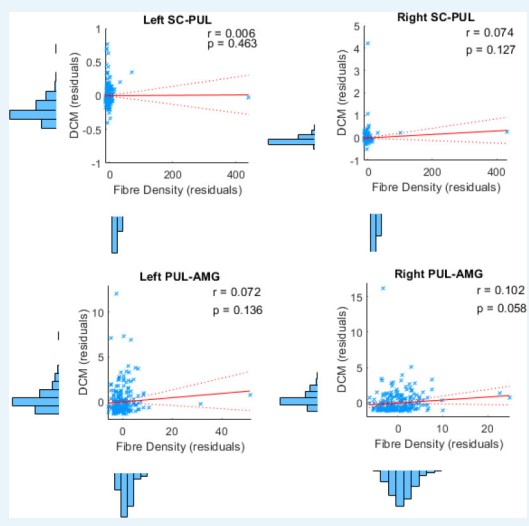

Local Tractography

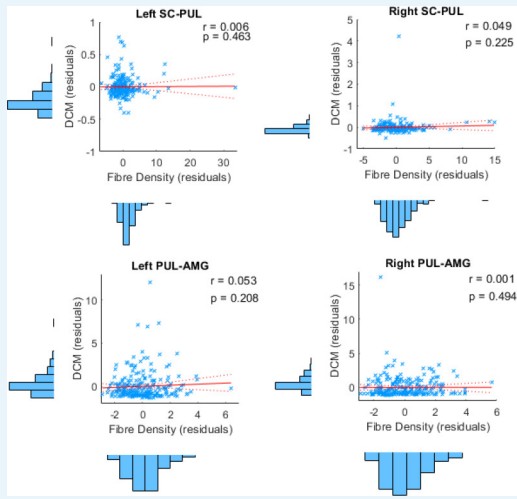

Local Tractography

DOI: https://doi.org/10.7554/eLife.40766.022

## Pulvinar and amygdala subregions

Repeated measures ANOVAs were conducted to compare the fibre counts (global tractography) or apparent fibre density (local tractography) on each pathway (SC-PUL, PUL-AMG) independently. Subregions (five clusters for the pulvinar, three subregions for the amygdala) and hemispheres (left, right) were compared. Absolute relative head motion was included as a covariate of no interest in each test. Greenhouse Geisser corrections made when Mauchly's test of sphericity was significant. All confidence intervals adjusted for Bonferroni correction. Follow-up paired *t*-tests conducted for significant interactions, bootstrapped with 1000 iterations. Outliers were removed according to whether participants had data on at least one variable with a standardised residual score above or below 3.

## Null tractography comparison

We conducted a series of paired *t*-tests between the number of streamlines generated by local probabilistic tractography using the iFOD2 algorithm vs. the null distribution algorithm

**Appendix 1—table 13.** Significance of DCM parameter estimates.

| Outliers removed | Parameter | M | SD | Df | T | P | 95% |
|---|---|---|---|---|---|---|---|
| | * Left SC-PUL | 1.063 | 0.115 | 229 | 8.285 | <0.001 | [1.048, 1.078] |
| | * Right SC-PUL | 1.059 | 0.124 | 233 | 7.295 | <0.001 | [1.043, 1.075] |
| | * Left PUL-AMG | 1.834 | 1.000 | 233 | 12.753 | <0.001 | [1.705, 1.962] |
| | * Right PUL-AMG | 1.731 | 0.765 | 230 | 14.514 | <0.001 | [1.632, 1.83] |
| | * Left IOG-FG | 1.685 | 0.715 | 228 | 14.485 | <0.001 | [1.592, 1.778] |
| | * Right IOG-FG | 2.415 | 2.140 | 233 | 10.111 | <0.001 | [2.139, 2.69] |
| | * Left FG-AMG | 1.213 | 0.258 | 233 | 12.614 | <0.001 | [1.18, 1.246] |
| | * Right FG-AMG | 1.190 | 0.217 | 230 | 13.308 | <0.001 | [1.162, 1.218] |
| | Left FG-IOG | 0.988 | 0.211 | 230 | −0.842 | 0.401 | [0.961, 1.016] |
| | Right FG-IOG | 1.037 | 0.270 | 231 | 2.095 | 0.037 | [1.002, 1.072] |
| | * Left AMG-FG | 1.813 | 1.330 | 231 | 9.306 | <0.001 | [1.641, 1.985] |
| | * Right AMG-FG | 1.878 | 1.101 | 228 | 12.078 | <0.001 | [1.735, 2.022] |

*Appendix 1—table 13 continued on next page*

| | Parameter | M | SD | df | t | p | 95% CI |
|---|---|---|---|---|---|---|---|
| No Outliers Removed | * Left SC-PUL | 1.066 | 0.157 | 236 | 6.481 | <0.001 | [1.046, 1.086] |
| | * Right SC-PUL | 1.068 | 0.190 | 236 | 5.510 | <0.001 | [1.044, 1.093] |
| | * Left PUL-AMG | 1.916 | 1.235 | 236 | 11.415 | <0.001 | [1.758, 2.074] |
| | * Right PUL-AMG | 1.821 | 0.944 | 236 | 13.382 | <0.001 | [1.7, 1.942] |
| | * Left IOG-FG | 1.864 | 1.214 | 236 | 10.957 | <0.001 | [1.709, 2.02] |
| | * Right IOG-FG | 2.772 | 4.249 | 236 | 6.422 | <0.001 | [2.229, 3.316] |
| | * Left FG-AMG | 1.251 | 0.446 | 236 | 8.661 | <0.001 | [1.194, 1.308] |
| | * Right FG-AMG | 1.218 | 0.278 | 236 | 12.081 | <0.001 | [1.182, 1.253] |
| | Left FG-IOG | 1.028 | 0.329 | 236 | 1.311 | 0.191 | [0.986, 1.07] |
| | Right FG-IOG | 1.069 | 0.348 | 236 | 3.071 | 0.002 | [1.025, 1.114] |
| | * Left AMG-FG | 2.057 | 2.166 | 236 | 7.517 | <0.001 | [1.78, 2.335] |
| | * Right AMG-FG | 2.092 | 1.596 | 236 | 10.536 | <0.001 | [1.888, 2.296] |

*p <.001

DOI: https://doi.org/10.7554/eLife.40766.023

implemented in MRtrix 3. Outliers with a difference more than three standard deviations from the mean were removed from the 622-participant sample for each paired *t*-test. Bootstrapping was set to 1000 iterations.

### Multivariate diffusion-behaviour relationships

We conducted two separate multivariate analyses of covariance, one for global and one for local tractography measures of fibres, to examine the relationship between emotional expression recognition (fearful, sad, and angry expressions) and fibre density along the subcortical route.

### Partial correlations between fibre density and effective connectivity parameters

We conducted a series of right-sided Pearson's partial correlations between the fibre density of each connection (left and right SC-PUL and PUL-AMG connections, global and local tractography – giving eight in total) and its corresponding DCM parameter estimate (modulatory effect of Faces > Shapes over region coupling). Head motion was entered as a control variable. Multivariate outliers were detected according to Mahalanobis distance ($df = 8$, $\chi^2$ criterion = 15.507, p=0.05), resulting in 24 participants being excluded from the analysis (N = 213). Note that outliers were substantially influencing the results (see table below for comparison).

### DCM parameter estimates

We conducted one-sample *t*-tests on the exponentiated DCM parameter estimates (B matrix) against a test value of 1 to examine whether modulatory connection strength was consistently greater than the prior within our sample of participants. Outliers were removed that were more than 3 SDs from the mean of each variable (note that this did not change the pattern of results).

### VOI inclusion/exclusion criteria

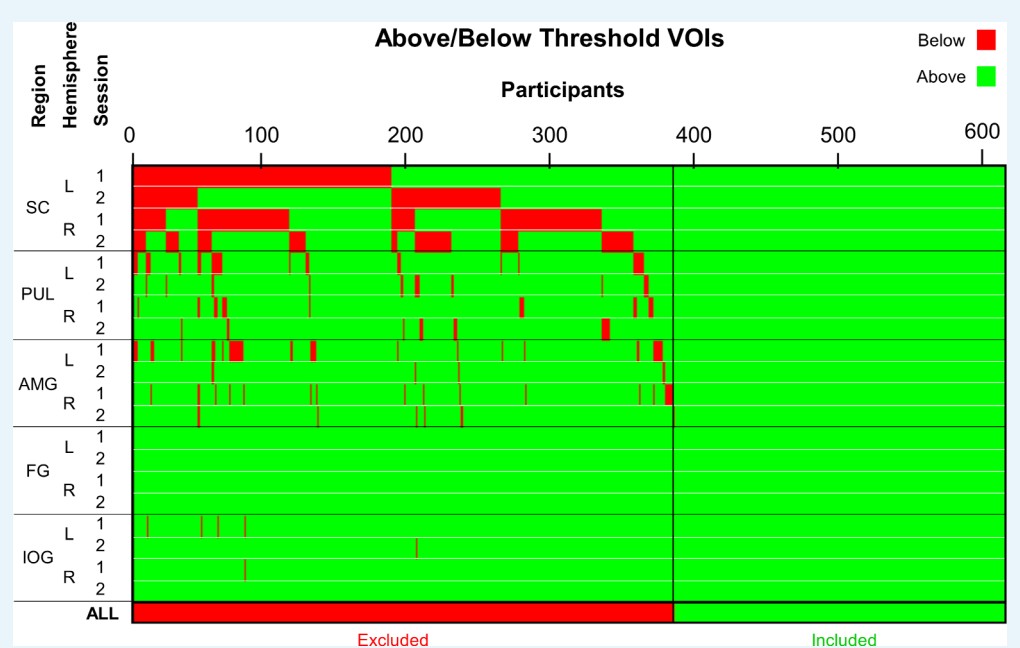

**Appendix 1—figure 1.** Participants with above or below threshold fMRI signal in all VOIs for inclusion into DCM stage. Threshold was set at p<0.05 uncorrected, using the [Faces – Shapes] contrast. Each column is a participant (N = 622) and each row is a different region (SC = superior colliculus, PUL = pulvinar, AMG = amygdala, FG = fusiform gyrus, IOG = inferior occipital gyrus) with two sessions each (i.e. each fMRI run) per hemisphere (L = left, R = right). Red indicates below-threshold signal and green indicates above-threshold signal. The bottom row ('ALL') indicates whether the participants were included or excluded from further DCM analysis, based on whether they had any below-threshold VOIs.
DOI: https://doi.org/10.7554/eLife.40766.024

