## [Decision Letter]

[**Editorial note:** This article has been through an editorial process in which the authors decide how to respond to the issues raised during peer review. The Reviewing Editor's assessment is that minor issues remain unresolved.]

Thank you for submitting your article "An afferent white matter pathway from the pulvinar to the amygdala facilitates fear recognition" for consideration by *eLife*. Your article has been reviewed by three peer reviewers, including Timothy Verstynen as the Reviewing Editor and Reviewer #1, and the evaluation has been overseen by Timothy Behrens as the Senior Editor. The following individuals involved in review of your submission have also agreed to reveal their identity: Jean Vettel (Reviewer #2); Marco Tamietto (Reviewer #3).

The Reviewing Editor has highlighted the concerns that require revision and/or responses, and we have included the separate reviews below for your consideration. If you have any questions, please do not hesitate to contact us.

As you can see from the reviews below, all three reviewers were very positive about this work and supportive of it being published. The analysis was smart, rigorous, and clearly theoretically motivated. The links to previous work were strong and yet the results were also clearly novel, making the work both incremental and novel (not an easy feat to pull off).

Separate reviews (please respond to each point):

*Reviewer #1:*

This article by McFayden and colleagues reports a beautiful set of multimodal analyses, using the HCP sample, to explore the existence of a subcortical route to the amygdala from the colliculus, via the pulvinar nucleus of the thalamus. Using dMRI, the authors identify plausible white matter pathways that connect 1) the amygdala to the pulvinar, and 2) the pulvinar to the superior colliculus. They then show that, behaviorally, the degree of estimated white matter connectivity between the amygdala and pulvinar predicted individual differences in recognition of fearful faces in the Penn Recognition Test. Using the dMRI structural connectivity information, they then use DCM on fMRI data from the Faces task to evaluate a set of models for information flow along both cortical and subcortical pathways. The winning model, which substantially out performs the other models, includes both a direct and indirect route to the pulvinar that relays information to the amygdala. Finally, the authors show that the degree of effective network connectivity from the DCM analysis correlated with the degree of structural connectivity analysis.

All in all, I thought this was a phenomenal study. First and foremost, I thought this was a very strong theoretically motivated paper. In an era of exploratory data analysis on structural and functional imaging data, it is refreshing to see such a tightly motivated theoretical paper. The multimodal analysis was well executed and the results were clear.

I do have some suggestions for improving the paper.

1) The authors are essentially making a mediation argument without directly stating it: i.e., white matter connectivity mediates a link between effective functional connectivity of the network and behavioral performance in the Penn Recognition Task. But the analysis ends without evaluating a formal mediation statistical analyses (e.g., Preacher and Hayes, 2008). Showing that white matter connectivity statistically mediates an indirect pathway between effective network connectivity and behavior would provide a powerful direct test of the links the authors are alluding to in the paper.

2) Since all of the results here are cross-sectional, the authors fail to account for a critical confound measure that can impact both dMRI and fMRI associations: head motion. The authors report that they include "head motion" as a regressor in the fMRI analyses, but do not describe what terms were used (see Bright et al,. 2017, Cleaning up the fMRI time series: Mitigating noise with advanced acquisition and correction strategies. NeuroImage, 154, 1-3.). In addition, no such control was applied to the associations with the dMRI connectivity (see Baum et al., 2018). The impact of in-scanner head motion on structural connectivity derived from diffusion MRI. Neuroimage, 173, 275-286.). This should be controlled for before making inferences about the between-subject associations.

3) The spatial resolution of the analysis, particularly the tractography analysis, seems underwhelming. For example, in Figure 1 it looks as if the authors are using a full thalamus ROI (although in Figure 6, it appears that they are using a selective pulvinar ROI). Are we sure these tracts are ending in the pulvinar? Also, where in the amygdala are the fibers terminating (and does this match previous neuroanatomy studies)?

4) Along the same lines as my previous point, there isn't much match to previous known neuroanatomy of white matter pathways. Do the paths of the PUL-AMG and SC-PUL connections follow specific white matter fascicles? If so, which fascicles and do the trajectories make sense based on anatomical tracing studies?

5) I am a little confused as to why the authors restricted the DCM analyses to the subset of 237 subjects that had significant within-subject first level GLM maps. In the Materials and methods, the authors report that the use of the subsample doesn't impact the results (third paragraph of subsection “Dynamic causal modelling”). Why was the sub-sampling done? What does it buy you?

6) The structural connectivity and behavior correlations (Figure 2) are all significant with the local tractography dMRI results. However, the effective connectivity and structural connectivity correlation (Figure 6) is significant with the global tractography results. The inconsistency in white matter connectivity measures in the associations is puzzling and somewhat worrisome that some of these correlations may be spurious.

7) In the DCM analysis, it seems that the null hypothesis is that there is not a meaningful subcortical route (either input-to-pulvinar or input-to-superior colliculus) that indirectly conveys information to the amygdala. But it doesn't appear that such a model was tested. Not that I expect this to be particularly problematic given how robust the model selection results are (Figure 4B). It just seems that this model is the most straight forward.

8) This paper uses pretty standard associative tests for the key results. However, given the large sample size, it should be possible to use cross-validation to assess whether variability along one dimension (e.g., structural connectivity) predicts another dimension (e.g., Penn Recognition Test scores). Hold out set prediction accuracy is more informative than simple association tests and should be used as often as possible.

Minor Comments:

– Introduction section, third paragraph: Given how many studies use the HCP data set, this isn't necessarily an "unprecedentedly large" sample.

– Introduction section and elsewhere: The authors repeatedly use the term "neural activity". But they are measuring hemodynamic activity.

– Subsection “Greater fibre density predicts better fearful face perception” and elsewhere: Please show all the statistical results, not just those that emerged as passing the significance threshold. The information on the effect sizes of all analyses are very meaningful, even if the results don't pass the null hypothesis test.

– Subsection “A forwards-only subcortical route is engaged during face processing” vs subsection “Greater fibre density relates to stronger effective connectivity”: Why are you using different outlier tests for different analyses? You should be consistent in your outlier test method.

*Reviewer #2:*

In the paper submitted by McFadyen and colleagues, the research examines the existence of a direct subcortical pathway for detection of fear from visual input. This question is strongly motivated from (1) animal models that have inspired a search for a human homolog, (2) lesion data from humans who can see affective stimuli without an intact V1, and (3) a rich review of research that has identified suggestive evidence of portions of the subcortical pathway but nothing cohesive to rule out alternatives.

The core strength of the work arose from exploiting the HCP dataset as a resource developed to enable a first pass big data approach, and then combining this large sample with a multimodal technical approach to examine structure (two methods), behavior (out of scanner task), and function (functional localizer and then directed connectivity modeled with 120 models). A noteworthy decision was the data-driven selection of functional regions from the face-shape task. The analytic flow is clearly communicated, results well illustrated and described, and the core interpretations largely justified.

The short list of suggestions for improvement include:

1) The description of the parameters for the DCM modeling is much too sparse to really understand or intuit the core features of the data variability and how it would have influenced performance of the 120 models. Suggest expanding details in the “Dynamic causal modelling “subsection of the main and then add more in the supplement.

2) Related, the text describing the above-threshold participants for the model was confusing (main subsection “Dynamic causal modelling”). The lines list the lowest number of above-threshold participants as 69 and 58 and then 4 and 3, yet the text states that all 237 participants had above-threshold in all ROIs. Why wouldn't 237 + [69 or 58 or 4 or 3] be the lowest number? The intuition of what is being communicated here about the HCP sample must be incorrect on my end, but alternative guesses have not panned out either.

3) Great job checking the difference between the 237 and 385 participants for all of the features listed in subsection “Dynamic causal modelling”, but can some simple stats be added to describe the quantitative element of this. Reveal anything trending?

4) Across all of the tests in the paper, was there any family-wise error applied to cover the sheer volume of comparisons?

5) It would be interesting to know whether the structural connections among the functional regions from the shape task (FG and IOG) could also account for differential performance on the face task. While this is not critical for the specific focus of this paper on subcortical fear pathway, it could augment our understanding about whether the subcortical fear pathway provides a unique avenue for fear response or if it more redundant with the canonical regions involved in face processing.

Minor Comments:

1) The description of the behavioral task could be easier to discern by just stating, "To this end, we examined behavioral data from an out-of-scanner task, the Penn Emotion Recognition Task, that assessed a different component of face processing than the in-scanner task."

2) Throughout the paper, there are a few instances of missing spaces between words and a parentheses and a missing period in subsection “Regions of Interest”.

3) The Results section employs suggestive language (“likely present” –) while the Discussion is very strongly worded (“unequivocally supports” –; “study settles” –). Any chance for softening since all findings are at the limit of our current methods, so softening and adding some text about methodological considerations would strengthen the paper's ability to impact future work.

Additional data files and statistical comments:

Additional document was clear and all is stated to be on GitHub.

*Reviewer #3:*

The paper by McFadyen and colleagues deals with an extremely relevant and hotly debated topic in neuroscience; namely, the anatomical existence, functional significance and behavioral impact of a subcortical direct pathway to human amygdala for fear perception.

Although there were previous empirical data and theoretical accounts in humans and non-human models supporting this contention, the present study presents several important advancements that, IMHO, deserve attention and publication in *eLife*.

Briefly, the present study marks a seminal departure from previous investigations that adopted a unimodal approach to focus either on anatomical or functional aspects only, and typically used small samples. Here data from HCP about more than 600 subjects are analyzed with state-of-the-art neuroimaging techniques. Evidence of anatomical existence of fiber tracts connecting SC, PULV and AMG are gathered using in-vivo tractography using both global tractography as well as more traditional ROI-based streamline probabilistic analysis. Noteworthy, results not only indicate that such connections can be reconstructed in a large sample, but also that the part of the PUVL involved is the inferior-lateral (i.e., visual) part of the PULV, as previously found in small samples of human and non-human primates. Next, the behavioral relevance of this pathway is demonstrated, correlating fiber density in these tracts with fear recognition. Then, the functional role and directionality is evaluated with DCM on fMRI data. Results show that the best model incorporates both cortical AND subcortical pathway to the AMG, and that the directionality of the latter is feedforward from SC and PULV to AMG. Finally, greater fiber density is related positively to effective connectivity in this subcortical pathway, therefore supporting further the link between its anatomical and functional properties.

I only have very minor comments ("minimal", I would say), to a paper already excellent.

Minor Comments:

I have only very minimal comments and suggestions that I would like the authors to take into consideration.

First, it is interesting to note that across the different analyses there is a common trend about lateralization on this subcortical pathway in the right hemisphere. Originally, laterality in fMRI response in the amygdala and subcortical structures to masked facial expressions was reported in Morris et al., 1998 Nature and interpreted as supporting the role of the right hemisphere in non-conscious emotion perception. Anatomically, a similar trend in DTI data was reported in Tamietto et al., 2012 that, however, did not reach standard statistical threshold. I would invite the authors to comment a bit more on this laterality effectin relation to previous works and neuropsychological evidence.

Second, there is recent and converting evidence in patients with V1 damage and "affective blindsight" that this subcortical pathway seems tuned to process low-spatial frequency information (e.g., for a recent paper Burra et al., 2018). Evidence in healthy participants seems more controversial, with some findings supporting a similar specialization for low spatial frequencies (e.g., Méndez-Bertolo et al., 2016, Carretié et al., 2017 JoCN), and other data, eminently from the same authors purporting a more "generalized" role (McFadyen et al., 2017). Possibly, some further speculation and more explicit reference to this topic would be of interest in the light of the new data.

Third, one strong hypothesis verified in non-human models is that the subcortical route to the AMG is composed by a functionally integrated disynaptic pathway whereby the superficial layers of the SC project to infero-later PULV and the same neurons in the PULV then send efferents to the lateral AMG (a summary of evidence about different subcortical or non-canonical pathway to the amygdala in humans and non human animals can be found in Diano et al., 2017 Front Psychol). In an attempt to provide indirect evidence in humans, some previous tractography studies, after tracing fiber connection between SC and PULV, and between PULV and AMG, also tried to verify whether a subsample of these reconstructed fibers could be considered as belonging to the same streamlines. If I am not missing something, the present analyses do not investigate this aspect. It would be nice to have this info if possible. I acknowledge the paper is already methodologically rich, so I am not explicitly asking additional analysis, even though this should not be particularly long to perform. However, mentioning to this aspect for future development would be appropriate.

There is a factual error in the Introduction section where authors say about Tamietto et al., 2012 paper that "The white matter structure of the subcortical route was estimated for the patient and for six healthy, age-matched controls". In fact, the healthy controls analyzed were 10, not 6.

---

## [Author Response]

Reviewer #1:

[…] All in all, I thought this was a phenomenal study. First and foremost, I thought this was a very strong theoretically motivated paper. In an era of exploratory data analysis on structural and functional imaging data, it is refreshing to see such a tightly motivated theoretical paper. The multimodal analysis was well executed and the results were clear.I do have some suggestions for improving the paper.1) The authors are essentially making a mediation argument without directly stating it: i.e., white matter connectivity mediates a link between effective functional connectivity of the network and behavioral performance in the Penn Recognition Task. But the analysis ends without evaluating a formal mediation statistical analyses (e.g., Preacher and Hayes, 2008). Showing that white matter connectivity statistically mediates an indirect pathway between effective network connectivity and behavior would provide a powerful direct test of the links the authors are alluding to in the paper.

Thank you for this suggestion. We agree that this would be an interesting analysis to do, given the biological plausibility that denser structural connectivity between the pulvinar and amygdala may partly enhance fearful face recognition because it facilitates stronger effective connectivity during face processing (i.e. independent variable = fibre density, dependent variable = fearful face recognition accuracy, mediator = effective connectivity). We believe, however, that this question would be better addressed in a future experiment that is more appropriately set up to do so. Some obstacles to this analysis in our manuscript include:

1) Task differences. The fMRI task was a matching task for faces (angry and fearful) and shapes, allowing us to observe modulation by face- (vs. shape-) related processing along the subcortical route. The behavioural task, on the other hand, was an explicit emotion recognition task. Given these task differences, we did not have a priori hypotheses that the DCM parameter estimates would significantly relate to the scores in the Penn Emotion Recognition Task.

2) Discrepancies between global and local tractography. Our results indicated that the fibre measures from the global tractography were significantly related to effective connectivity, while the local tractography was significantly related to the behavioural performance to fearful faces. Hence, we would not expect either of these measures on their own to produce a significant mediation model including all three components.

We hope that we have explained our position clearly. We’re intrigued by this idea and would be keen to incorporate this approach in future studies.

2) Since all of the results here are cross-sectional, the authors fail to account for a critical confound measure that can impact both dMRI and fMRI associations: head motion. The authors report that they include "head motion" as a regressor in the fMRI analyses, but do not describe what terms were used (see Bright et al., 2017, Cleaning up the fMRI time series: Mitigating noise with advanced acquisition and correction strategies. NeuroImage, 154, 1-3.). In addition, no such control was applied to the associations with the dMRI connectivity (see Baum et., 2018). The impact of in-scanner head motion on structural connectivity derived from diffusion MRI. Neuroimage, 173, 275-286.). This should be controlled for before making inferences about the between-subject associations.

Thank you for raising this point. Regarding the fMRI motion correction, we have updated the manuscript to more explicitly state the features of the motion regression in the first-level analysis:

“We partitioned the GLM into sessions (left-to-right and right-to-left encoding) and we included 12 head motion parameters as multiple regressors (6 estimates from rigid-body transformation, and their temporal derivatives).”

Thank you for drawing our attention to this paper on dMRI and motion correction. We agree that this is an important confound to consider, and so we have now included head motion in all our dMRI analyses. See the new addition to the Materials and methods section below:

“For the diffusion images, this included intensity normalisation across runs, echo planar imaging (EPI) distortion correction and eddy current/motion correction (for full details, see Glasser et al., 2013). The latter stage produced motion parameters (three translations and three rotations), across which we computed the mean relative displacement between dMRI volumes. These values per participant were used in all further analyses to account for any confounding effect of motion (Baum et al., 2018). We conducted all further processing using MRTrix 3.0.15 (Tournier et al., 2012) and FSL 5.0.7.”

Importantly, including head motion as a covariate did not alter the pattern of our results. We have adjusted the “Reconstructing the subcortical route using tractography” Results section to update the statistical values:

“After covarying out head motion and removing four participants with outlying standardised residuals (*z*-score threshold = ± 3), we established that there were significantly more fibres connecting the SC and PUL (*M* = 13.119, *95% CI* = [12.738, 13.500]) than the PUL and AMG (*M* = 6.040, *95% CI* = [5.867, 6.214]; *F*(1,616) = 433.286, *p* = 2.842 × 10^-73^, η_p_^2^ =.413).We also found a hemispheric lateralisation, such that there were more reconstructed fibres for the right (*M* = 9.879, *95% CI* = [9.624, 10.135]) than the left (*M* = 9.280, *95% CI* = [9.023, 9.537]) hemisphere (*F*(1,616) = 7.583, *p* =.006, η_p_^2^ =.012), specifically for the PUL-AMG pathway (*F*(1,616) = 16.025, *p* = 7.000 × 10^-5^, η_p_^2^ =.025; *t*(617) = -9.785, *p* = 4.070 × 10^-18^, 95% CI [-1.714, -1.141]).”

“The apparent fibre density of the streamlines generated using local tractography followed the same pattern as the global tractography fibre counts, such that there was greater fibre density for the SC-PUL connection (*M* = 5.793, *95% CI* = [5.663, 5.923]) than the PUL-AMG connection (*M* = 4.461, *95% CI* = [4.368, 4.554]; *F*(1,607) = 69.586, *p* = 4.930 × 10^-16^, η_p_^2^ =.103), after accounting for head motion and removing 13 outliers according to their residuals. Fibre density was also greater on the right (*M* = 4.935, *95% CI* = [4.822, 5.048]) than the left (*M* = 3.987, *95% CI* = [3.889, 4.086]) for the PUL-AMG connection (*t*(608) = -18.205, *p* = 1.960 × 10^-59^, 95% CI [-1.050, -0.845]) while, in contrast, there was greater fibre density for the left than right SC-PUL connection (*t*(608) = 10.749, *p* = 8.600 x 10^-25^, *95% CI* = [0.742, 1.073]; *F*(1,607) = 162.475, *p* = 3.828 × 10^-33^, η_p_^2^ =.211). Taken together, our tractography analyses provide strong convergent evidence for a subcortical white matter pathway to the amygdala in the human brain.”

We have also adjusted the “Greater fibre density predicts better fearful face perception” Results section:

“Thus, we entered fibre density measures for the SC-PUL and PUL-AMG pathways into two separate multivariate regressions (one per tractography method, to reduce collinearity) with recognition accuracy scores for fearful, angry, and sad faces as covariates, plus head motion as a control covariate. We removed outliers (4 for global tractography, 15 for local tractography) with residuals ± 3 standard deviations from the mean. While there were no significant multivariate relationships between global tractography and emotion recognition (see Supplementary Materials, Tables 10 and 11 for detailed statistics), there was a significant relationship between local tractography and recognition accuracy for fearful faces (*F*(4,598) = 2.501, *p* =.042, Wilk’s Λ =.984, n_p_^2^ =.016; see Figure 2B). This was driven predominantly by fibre density of the left (*β* = 0.140, *p* =.004) and right (*β* = 0.143, *p* =.012) PUL-AMG connections’ local fibre density. The local fibre density of the left and right SC-PUL did not contribute significantly to the model. These results suggest that the fibre density of the PUL-AMG half of the subcortical route is associated with fearful face recognition more so than with other negative (sad) or threatening (angry) emotional expressions.”

And also the “Greater fibre density relates to stronger effective connectivity” Results section:

“We computed eight partial correlations (with head motion as a control covariate) to examine the relationship between each parameter estimate and the corresponding global fibre count and local summed weights per connection (left and right SC-PUL and PUL-AMG). After removing multivariate outliers (leaving N = 213; see Supplementary Materials, Table 12 for details), we discovered that participants with more global fibres also had greater modulatory activity for the right (*r* =.180, *p* =.004, *p_bonf_* =.032; see Figure 6) but not the left (*r* =.095, *p* =.084, *p_bonf_* =.672) PUL-AMG connection. The SC-PUL connection was not significantly related to its corresponding DCM parameters for global (left: *r* = -.022, *p* =.627, *p_bonf_* = 1.000; right: *r* =.002, *p* =.488, *p_bonf_* = 1.000) or local (left: *r* = -0.101, *p* =.928, *p_bonf_* = 1.000; right: *r* = -.028, *p* =.659, *p_bonf_* = 1.000) tractography.”

We have also updated the data points shown in Figure 2 and Figure 6 so that the predicted fibre densities reflect the multivariate analysis with head motion included as a covariate.

3) The spatial resolution of the analysis, particularly the tractography analysis, seems underwhelming. For example, in Figure 1 it looks as if the authors are using a full thalamus ROI (although in Figure 6, it appears that they are using a selective pulvinar ROI). Are we sure these tracts are ending in the pulvinar? Also, where in the amygdala are the fibers terminating (and does this match previous neuroanatomy studies)?

The resolution of the images provided by the HCP are: 1.25 mm isotropic voxels for the dMRI, 2 mm isotropic voxels for the fMRI, and 0.7 mm isotropic voxels for the MRI. For our tractographic reconstruction, we optimised our parameters according to MRTrix’s recommendations for the high-resolution HCP images (https://mrtrix.readthedocs.io). Given these spatial resolutions, we are confident that our ROI masks for seeding and terminating tracks accurately reflect the location of both the pulvinar and amygdala.

The selective pulvinar ROI depicted in Figures 1 and 6 was generated by an fMRI meta-analysis at a resolution of 2 mm isotropic (Barron et al., 2015). We did not use a full thalamus ROI at any stage in the manuscript, given that we were only interested in the pulvinar (at particular exclusion to the nearby LGN).

Regarding where the tracts terminate in the pulvinar and amygdala, we were intrigued by this comment and have conducted further analyses to investigate. We created maps of where each streamline terminated, for both global and local tractography, and summed this value within subregion-specific amygdala and pulvinar masks. Details of the new amygdala subregions have been included in the Regions of Interest methods section:

“For the amygdala (AMG) binary mask, we used the probabilistic Harvard-Oxford Subcortical atlas at a threshold of at least 50% probability. For amygdala subregions, we used the basolateral, centromedial, and superficial amygdala regions in the Juelich Histological Atlas (Amunts et al., 2005) at a threshold of at least 40% probability.”

The manuscript has been updated to include these new results:

“To uncover more anatomical features of the reconstructed fibres, we used subregion-specific masks of the amygdala (basolateral, centromedial, and superficial) and the pulvinar (anterior, medial, superior, inferior, and lateral; see Materials and methods for ROI specification details) to determine where the reconstructed fibres terminated. This masking approach revealed that the global tractography fibres present between the SC and PUL connected predominantly to the inferior and anterior pulvinar (see Supplementary Materials Tables 3 to 5 for detailed statistics). Between the PUL and AMG, fibres terminated almost exclusively in the inferior PUL and then predominantly in the basolateral AMG. Hence, the inferior pulvinar served as the connecting node between the SC and the basolateral AMG for the globally-reconstructed subcortical pathway.”

“Like in the global tractography, we investigated the overlap between the locally-generated tracks and known white matter fasciculi. The pattern of results was the same, with up to 60% of fibres traversing the anterior thalamic radiation, corticospinal tract, and inferior longitudinal and fronto-occipital fasciculi. For the SC-PUL pathway, the majority of track density was found in the anterior thalamic radiation (left: *M* = 52.85%, *SD* = 11.70%, range = 16.55% to 96.96%; right: *M* = 58.21%, *SD* = 16.24%, range = 5.49% to 89.30%) and the corticospinal tract (left: *M* = 12.89%, *SD* = 8.78%, range = 0.18% to 37.84%; right: *M* = 32.35%, *SD* = 12.49%, range = 0.56% to 63.64%; see Figure 1). For the PUL-AMG pathway, the majority was found in the corticospinal tract (left: *M* = 32.29%, *SD* = 15.58%, range = 0.59% to 65.25%; right: *M* = 37.38%, *SD* = 16.21%, range = 0.32% to 65.99%), followed by the anterior thalamic radiation (left: *M* =16.47%, *SD* = 9.20%, range = 5.24% to 59.52%; right: *M* = 7.00%, *SD* = 3.49%, range = 1.46% to 50.09%), and then the inferior longitudinal (left: *M* = 5.69%, *SD* = 7.92%, range = 0% to 50.75%; right: *M* = 3.96%, *SD* = 6.33%, range = 0% to 47.51%) and fronto-occipital (left: *M* = 1.54%, *SD* = 4.21%, range = 0% to 72.79%; right: *M* = 7.96%, *SD* = 10.99%, range = 0% to 79.60%) fasciculi. Mean track densities were lower than 0.20% and 0.01% in other fasciculi for SC-PUL and PUL-AMG, respectively. We also examined which subregions of the pulvinar and amygdala the seeded probabilistic tracks terminated in. For the SC-PUL pathway, the greatest number of streamlines terminated in the anterior PUL, followed by the inferior pulvinar (see Supplementary Materials, Tables 6 to 8 for detailed statistics), consistent with the global tractography. Also like the global tractography, the local tractography fibres between the PUL and AMG terminated almost exclusively in the inferior PUL. For the AMG, however, fibres terminated predominantly in the basolateral subregion in the left hemisphere (consistent with the global tractography) but in the centromedial amygdala on the right.”

We have updated Figure 1 to illustrate these new findings.

We have also added a new paragraph to the Discussion to relate these subregion-specific findings to the literature:

“Our investigation into pulvinar and amygdala subregions support these findings, such that we found the superior colliculus to project predominantly onto the inferior (and anterior) pulvinar, which was the same subregion to receive the vast majority of fibres from the amygdala (see Figure 1). Furthermore, pulvinar fibres terminated predominantly within the basolateral amygdala, which is known to process visual information about threat and faces (Hortensius et al., 2016). Further studies could use both anatomically-constrained tractography and this subregion-specific approach with ultra-high-resolution imaging to better differentiate grey-white matter boundaries and more accurately determine if and where a continuous, subcortical route might traverse the pulvinar.”

Also, please note that we have decided to remove the “Assessing the confluence of the subcortical route by parcellating the pulvinar” from the Supplementary Materials (references to this in the main text have also been removed) as we feel that this new subregion-specific analysis added to the main text adds more meaningful information.

4) Along the same lines as my previous point, there isn't much match to previous known neuroanatomy of white matter pathways. Do the paths of the PUL-AMG and SC-PUL connections follow specific white matter fascicles? If so, which fascicles and do the trajectories make sense based on anatomical tracing studies?

Thank you for this great suggestion. We used the Johns Hopkins University DTI-based white matter atlas implemented in FSL to determine which major fasciculi overlapped with our generated fibres. We have added this new analysis to the Results section:

“We used the probabilistic JHU DTI-based white matter atlas (Mori et al., 2009), implemented in FSL, to examine any overlap between 20 major fasciculi and the globally-reconstructed fibres. After warping the tractograms into standard space and converting them into track density images, we calculated the total fibre density within each fasciculus. This revealed that up to 60% of the subcortical route overlapped with major fasciculi, mainly the anterior thalamic radiation and the corticospinal tract, as well as the inferior longitudinal and fronto-occipital fasciculi. For the SC-PUL pathway, the major overlap was found in the anterior thalamic radiation in the left (*M* = 56.11%, *SD* = 15.56%, range = 9.46% to 100%) and right (*M* = 55.78%, *SD* = 16.97%, range = 4.81% to 96.55%) hemispheres, followed by the corticospinal tract (left: *M* = 4.82%, *SD* = 6.08%, range = 0% to 33%; right: *M* = 21.06%, *SD* = 12.75%, range = 0% to 69.09%; see Figure 1). All other fasciculi had mean track densities less than 0.06% of the full SC-PUL pathway. Track density of the PUL-AMG pathway was mostly found in the corticospinal tract (left: *M* = 33.69%, *SD* = 18.56%, range = 0% to 88.89%; right: *M* = 36.73%, *SD* = 17.80%, range = 0% to 82.35%), followed by the anterior thalamic radiation (left: *M* = 20.58%, *SD* = 14.10%, range = 0% to 70%; right: *M* = 11.32%, *SD* = 8.88%, range = 0% to 52.75%). There was also some overlap with the inferior longitudinal fasciculus (left: *M* = 5.44%, *SD* = 8.60%, range = 0 to 65.52%; right: *M* = 3.46%, *SD* = 5.63%, range = 0 to 40.68%) and the inferior fronto-occipital fasciculus (left: *M* = 1.49%, *SD* = 6.73%, range = 0 to 86.96%; right: *M* = 7.24%, *SD* = 10.33%, range = 0 to 75.72%). Mean track densities in all other fasciculi were less than 0.30%.”

“Like in the global tractography, we investigated the overlap between the locally-generated tracks and known white matter fasciculi. The pattern of results was the same, with up to 60% of fibres traversing the anterior thalamic radiation, corticospinal tract, and inferior longitudinal and fronto-occipital fasciculi. For the SC-PUL pathway, the majority of track density was found in the anterior thalamic radiation (left: *M* = 52.85%, *SD* = 11.70%, range = 16.55% to 96.96%; right: *M* = 58.21%, *SD* = 16.24%, range = 5.49% to 89.30%) and the corticospinal tract (left: *M* = 12.89%, *SD* = 8.78%, range = 0.18% to 37.84%; right: *M* = 32.35%, *SD* = 12.49%, range = 0.56% to 63.64%; see Figure 1). For the PUL-AMG pathway, the majority was found in the corticospinal tract (left: *M* = 32.29%, *SD* = 15.58%, range = 0.59% to 65.25%; right: *M* = 37.38%, *SD* = 16.21%, range = 0.32% to 65.99%), followed by the anterior thalamic radiation (left: *M* =16.47%, *SD* = 9.20%, range = 5.24% to 59.52%; right: *M* = 7.00%, *SD* = 3.49%, range = 1.46% to 50.09%), and then the inferior longitudinal (left: *M* = 5.69%, *SD* = 7.92%, range = 0% to 50.75%; right: *M* = 3.96%, *SD* = 6.33%, range = 0% to 47.51%) and fronto-occipital (left: *M* = 1.54%, *SD* = 4.21%, range = 0% to 72.79%; right: *M* = 7.96%, *SD* = 10.99%, range = 0% to 79.60%) fasciculi. Mean track densities were lower than 0.20% and 0.01% in other fasciculi for SC-PUL and PUL-AMG, respectively.”

We have also updated Figure 1 to visualise these findings (see previous comment response).

5) I am a little confused as to why the authors restricted the DCM analyses to the subset of 237 subjects that had significant within-subject first level GLM maps. In the Materials and methods, the authors report that the use of the subsample doesn't impact the results (third paragraph of subsection “Dynamic causal modelling”). Why was the sub-sampling done? What does it buy you?

The reason for restricting the DCM analysis to a subsample of 237 participants was that these were the participants who had useable data – that is, who had above-threshold (*p* <.05) signal in the superior colliculus. As the superior colliculus was included in all our hypothesised models, it was essential that participants had sufficient signal (for Faces > Shapes). As Stephan et al., 2009, explains, “Indeed, it would be nonsensical to ask this question of regional responses that did not show experimental effects. Generally, one should use the most revealing *t*- or *F*-contrast for each region, to identify the local maxima in subject-specific SPMs that are nearest to the maximum in the group SPM,” (Stephan et al., 2009, “Ten simple rules for dynamic causal modeling”). For 385 participants, we were not able to identify a reliable local maxima, as there were no voxels below *p* <.05 uncorrected in the Faces > Shapes contrast. For clarity, we have now included a table of participants who met this criteria per ROI in the Supplementary Materials: Figure 4 —figure supplement 1

6) The structural connectivity and behavior correlations (Figure 2) are all significant with the local tractography dMRI results. However, the effective connectivity and structural connectivity correlation (Figure 6) is significant with the global tractography results. The inconsistency in white matter connectivity measures in the associations is puzzling and somewhat worrisome that some of these correlations may be spurious.

This is a great point and one that we have deliberated over. Unfortunately, we don’t have a definitive answer on this one, but here are some of our thoughts on the matter:

a) While there is indeed a discrepancy, which you have pointed out, there are also many consistencies between the two methods. This is most evident by the pattern of results for the relative fibre densities between SC-PUL and PUL-AMG and their subregions (Figure 1), as well as the similar overlap with major fasciculi.

b) It does seem, however, that the streamlines generated by global tractography are more conservative (i.e. fewer in number) and follow a more direct path between ROIs (see example subject shown in Figure 1). It has, indeed, been established that global tractography is more effective in modelling local connection architecture (Jbabdi and Johansen-Berg, 2011). Additionally, we observed that head motion often significantly covaried with local, but not global, tractography (see Supplementary Materials, Tables 1 to 8). This suggests that the local tractography may indeed be more susceptible to noise, inspiring relatively more confidence in the global tractography/DCM relationship than perhaps the local tractography/behaviour relationship. That said, the relationship we observed between the local tractography and the emotion recognition scores was as we hypothesised and was assessed using a full multivariate test.

We encourage future studies to explicitly investigate discrepancies between local and global tractography methods (for example, see Anastasopoulos et al., 2014, Local and Global Fiber Tractography in Patients in Epilepsy), particularly pertaining to tractography-function or tractography-behaviour relationships.

We have now included these points in the Discussion:

“One limitation of the present study is the discrepancy between how local and global measures of fibre density related to other measures; namely, that local tractography covaried with fearful face recognition scores while global tractography covaried with effective connectivity. While the reconstructed fibres shared many similarities (e.g. the pattern of findings for each connection across hemispheres and subregions, as well as the overlap with major fasciculi; see Figure 1) even after accounting for head motion, it is possible that the local tractography’s relatively greater susceptibility to noise may have decreased its relationship to corresponding effective connectivity parameters. Indeed, global tractography has been shown to better reflect local connection architecture (Jbabdi and Johansen-Berg, 2011), such as the subcortical connections we have investigated. Such discrepancies between global and local tractography have been reported in other studies (Anastasopoulos et al., 2014) and so further research (particularly those that only recruit a single tractograpy method) will benefit from specific investigations into why these discrepancies might arise.”

7) In the DCM analysis, it seems that the null hypothesis is that there is not a meaningful subcortical route (either input-to-pulvinar or input-to-superior colliculus) that indirectly conveys information to the amygdala. But it doesn't appear that such a model was tested. Not that I expect this to be particularly problematic given how robust the model selection results are (Figure 4B). It just seems that this model is the most straight forward.

In the DCM analysis, the ‘null’ hypothesis was captured by models where there was not an effective connection between the superior colliculus, pulvinar, and amygdala (upper left of Figure 4A). We also tested the alternate hypothesis that a subcortical route indirectly conveys information to the amygdala (as raised in this comment) via the inferior occipital gyrus (Figure 4A: row 1 column 6) and fusiform gyrus (Figure 4A: row 1, column 6). If we understand the comment correctly, then the null hypothesis here would be tested by comparing the former (i.e. ‘cortical-only’ models) with the latter (i.e. ‘subcortical-cortical’ or ‘indirect’ models).

8) This paper uses pretty standard associative tests for the key results. However, given the large sample size, it should be possible to use cross-validation to assess whether variability along one dimension (e.g., structural connectivity) predicts another dimension (e.g., Penn Recognition Test scores). Hold out set prediction accuracy is more informative than simple association tests and should be used as often as possible.

Thank you for this suggestion. If we understand correctly, the suggestion is to conduct cross-validated linear regressions for the tractography-DCM and tractography-behaviour relationships. We agree that this would be an informative and potentially powerful analysis to conduct. However, given the magnitude of this dataset and our inexperience with this type of analysis, we are reluctant to re-analyse the tractography-DCM and tractography-behaviour relationships that we have already demonstrated using standard associative tests. We will certainly keep this suggestion in mind for future studies.

Minor Comments:– Introduction section, third paragraph: Given how many studies use the HCP data set, this isn't necessarily an "unprecedentedly large" sample.

Thank you for identifying this. Our intention was to convey that our sample size is the largest for a study reconstructing the subcortical route. We have removed “unprecedently” to avoid confusion.

“First, we used DWI to reconstruct the subcortical route from the superior colliculus to the amygdala, via the pulvinar, and estimated its fibre density on a large sample.”

– Introduction section and elsewhere: The authors repeatedly use the term "neural activity". But they are measuring hemodynamic activity.

Thank you for noting this. We have updated this throughout the manuscript where we make reference to BOLD/haemodynamic responses observed in the present fMRI experiment.

Abstract

“We then computationally modelled the flow of haemodynamic activity during a face-viewing task and found evidence for a functionally-afferent pulvinar-amygdala pathway”

Introduction

“Next, we modelled the direction-specific flow of haemodynamic responses to faces, testing whether a functional subcortical route is recruited to transmit information toward the amygdala.”

Results

“The winning model revealed that the functional network that best explained the BOLD responses in our sample of 237 participants included visual inputs to the superior colliculus and pulvinar, forward connections from superior colliculus to the amygdala via the pulvinar, and recurrent interactions between IOG and FG, as well as between FG and amygdala.”

– Subsection “Greater fibre density predicts better fearful face perception” and elsewhere: Please show all the statistical results, not just those that emerged as passing the significance threshold. The information on the effect sizes of all analyses are very meaningful, even if the results don't pass the null hypothesis test.

Thank you for this suggestion. Due to the large number of statistical tests in this manuscript, we have added 13 tables to the Supplementary Materials that thoroughly detail all computed statistical tests. References to the relevant tables have now been added to the main text to point readers towards the specific statistics of both non-significant and significant test results.

– Subsection “A forwards-only subcortical route is engaged during face processing” vs subsection “Greater fibre density relates to stronger effective connectivity”: Why are you using different outlier tests for different analyses? You should be consistent in your outlier test method.

Thank you for raising this point. We have revisited our method of outlier removal so that it is appropriate for the assumptions of each statistical test. In the case of repeated measures ANOVA (in the first Results section on the tractography) and the multivariate ANOVA (in the second Results section on diffusion-behaviour relationships), we now identify outliers by calculating the residuals of each variable and removing participants who had any z-scored residuals above or below 3 (we previously had identified outliers as participants whose data – not residuals – were more than 3 standard deviations from the mean). For the partial correlations, we have now identified multivariate outliers with a Mahalanobis distance over 15.507 (based on χ^2^ distribution with *df* of 8 at *p* <.05). For the one-sample *t*-tests of the DCM parameters against a test value of 1, we have kept our original outlier method of values more than 3 SDs from the mean as this is appropriate for one-sample *t*-tests (and also means we are likely excluding participant’s whose parameter estimates may reflect being stuck in a local minima/maxima). These details are now stated in each Results section. Also note that we report all statistics with *and* without outliers in the newly updated Supplementary Materials for comparison and to demonstrate that outlier removal made essentially no difference to the pattern of results.

Reviewer #2:

[…] The core strength of the work arose from exploiting the HCP dataset as a resource developed to enable a first pass big data approach, and then combining this large sample with a multimodal technical approach to examine structure (two methods), behavior (out of scanner task), and function (functional localizer and then directed connectivity modeled with 120 models). A noteworthy decision was the data-driven selection of functional regions from the face-shape task. The analytic flow is clearly communicated, results well illustrated and described, and the core interpretations largely justified.The short list of suggestions for improvement include:1) The description of the parameters for the DCM modeling is much too sparse to really understand or intuit the core features of the data variability and how it would have influenced performance of the 120 models. Suggest expanding details in the “Dynamic causal modelling “subsection of the main and then add more in the supplement.

Thank you, we realise now that we only mentioned the Bayesian Model Selection (i.e. the process by which we compare the 102 models) in the main text of the Results and not the Materials and methods section. We have now added this to the end of the dynamical causal modelling Results section:

“The final model space consisted of 102 models (see Figure 5), where the first Cortical family contained six models, the second and third Cortical families contained 12 models each, and the Dual families (families 4, 5, and 6) contained 24 models each. The different families correspond to different input types (superior colliculus only, pulvinar only, or superior colliculus and pulvinar) and the different models within these families arise from different combinations of forward and backward connections. Each of the final 102 DCMs were modelled separately for both fMRI sessions. Both hemispheres were included in each model with no cross-hemispheric connections. To determine which model best explained the data, we conducted family-wise Bayesian Model Selection (Stephan et al., 2009; Rigoux et al., 2014), which penalises models for complexity according to the free energy principle (Friston et al., 2006). We used the random effects implementation to account for potential individual differences in the recruitment of a subcortical pathway for viewing faces (Stephan et al., 2009).”

We would also like to note that the code used to specify the DCMs can be found here: https://github.com/jjmcfadyen/hcp-diffusion-dcm/blob/master/specify_DCM.m, as can the template DCM used for this script: https://github.com/jjmcfadyen/hcp-diffusion-dcm/blob/master/DCM_template.mat

2) Related, the text describing the above-threshold participants for the model was confusing (main subsection “Dynamic causal modelling”). The lines list the lowest number of above-threshold participants as 69 and 58 and then 4 and 3, yet the text states that all 237 participants had above-threshold in all ROIs. Why wouldn't 237 + [69 or 58 or 4 or 3] be the lowest number? The intuition of what is being communicated here about the HCP sample must be incorrect on my end, but alternative guesses have not panned out either.

Thank you for drawing our attention to this. We realised that we had mistyped and reported the incorrect numbers. We had meant to say:

“This was the case for 237 out of the 622 participants (see Figure 4 —figure supplement 1). The ROIs with the highest numbers of below-threshold participants were the left and right superior colliculi (261 and 246 participants, respectively), followed by the left and right pulvinar (46 and 32 participants, respectively)”

We have also now included a heatmap of which participants had above-threshold signal in which areas to the Supplementary Materials for full transparency (see response to reviewer 1 above).

3) Great job checking the difference between the 237 and 385 participants for all of the features listed in subsection “Dynamic causal modelling”, but can some simple stats be added to describe the quantitative element of this. Reveal anything trending?

Thank you for this suggestion. We have now reported the lowest *p* values of all ‘group’ (i.e. participants included or excluded from the DCM analysis) effects. For the global and local tractography, this was done by entering in ‘group’ as a covariate of interest. For all other variables, this was done by computing independent-samples *t*-tests. Notably, only one result was *p* <.1 (volume of left thalamus *p* =.055). Considering that these results were not corrected for multiple comparisons (critical Bonferroni value should be *p* <.002 for 19 independent-samples *t*-tests), none of these values were approaching significance.

*“*Critically, the group of 237 participants with above-threshold BOLD responses in all ROIs did not differ significantly from the other group of 385 participants in the global or local tractography results (main effect of ‘group’ and interactions with ‘group’ were all *p* >.162 and η_p_^2^ <.003), performance on the Penn Emotion Recognition task (all independent-samples *t*-tests had *p* >.100), volume of the thalamus (left: *p* =.055, right: *p* =.987) / amygdala (left: *p* =.472, right: *p* =.394) / fusiform area (left: *p* =.677, right: *p* =.597) / lateral occipital area (left: *p* =.762, right: *p* =.679; volumes computed by Freesurfer), median reaction time (faces: *p* =.418, shapes: *p* =.617) and accuracy (faces: *p* =.417, shapes: *p* =.717) during the fMRI task, age (*p* =.782), or gender (*p* =.359). Therefore, using the information available to us, we had no evidence to assume that our DCM sample was biased by any confounding variable.”

4) Across all of the tests in the paper, was there any family-wise error applied to cover the sheer volume of comparisons?

We agree that there were a very large number of statistical tests computed in this manuscript (as can now been clearly seen in the statistical tables we have added to the Supplementary Materials). Deciding what a ‘family’ of tests refers to is a highly contentious issue but, given that we had many *a priori* conceptually-different hypotheses, we decided that it would be appropriate to partition our statistical tests into families, rather than to apply experiment-wise correction. As such, we partitioned our multiple-comparison-correction into 4 different families for 4 different hypotheses. These hypotheses and the details of the corrections in each are listed below:

1) That the subcortical route could be reconstructed using both global and local tractography in all participants (1^st^ Results section: Reconstructing the subcortical route using tractography)

– Tests conducted under this hypothesis aimed to quantify anatomical features of each pathway (e.g. whether there was greater fibre density in the SC-PUL or PUL-AMG, left or right hemisphere, and to the different pulvinar and amygdala subregions).

– We have reported the original, uncorrected *p* values in the Supplementary Materials, Tables 1 to 9. At a critical *p* value of.005 (Bonferroni correction for 9 comparisons), all results remain the same except for the main effect of hemisphere for global tractography across SC-PUL and PUL-AMG connections.

2) That greater fibre density of the subcortical route would relate to better emotional expression recognition (2^nd^ Results section: Greater fibre density predicts better fearful face perception)

– We conducted two separate MANOVAs (one per tractography method) to reduce multicollinearity. In accordance with convention, we did not correct for these two comparisons. While it is true that a correction such as this would have altered the main multivariate effect of Fear Recognition (*p* =.042, *p_bonf_* =.084), it would not have altered the significance of the parameter estimates for left (*p* =.004, *p_bonf_* =.008) and right (*p* =.012, *p_bonf_* =.024) PUL-AMG fibre density on Fear Recognition.

3) That a functional, forwards-only subcortical route would be engaged during face processing (4^th^ Results section: A forwards-only subcortical route is engaged during face processing):

– This was tested by first conducting a GLM on the Faces vs. Shapes data (*p* <.05 FWE-corrected according to Random Field Theory in SPM12) and then by estimating and comparing dynamic causal models using Bayesian statistics.

– Comparisons between each parameter estimate and the prior were Bonferroni-corrected.

4) That stronger effective connectivity along the subcortical route would relate to greater fibre density (Results section: Greater fibre density relates to stronger effective connectivity)

– Comparisons between each pathway’s fibre density and effective connectivity estimate were Bonferroni-corrected.

5) It would be interesting to know whether the structural connections among the functional regions from the shape task (FG and IOG) could also account for differential performance on the face task. While this is not critical for the specific focus of this paper on subcortical fear pathway, it could augment our understanding about whether the subcortical fear pathway provides a unique avenue for fear response or if it more redundant with the canonical regions involved in face processing.

Thank you for this great comment. If we understand correctly, the suggestion is to compare the relative contribution of fibre density along cortical visual pathways with fibre density along the subcortical route to emotional face recognition. We have partially addressed this suggestion with the dynamic causal modelling, by showing that models including a subcortical route explained the data above and beyond the penalty of additional complexity. This indicates that the functional engagement of the subcortical route is not likely redundant when comparing face to shape processing.

Regarding specifically fear processing, it would certainly be interesting to see how cortical visual pathways compare to the subcortical route in predicting fearful face recognition. There are several potential structural connections between the amygdala and visual cortical areas that we could investigate, including the occipital cortex, temporal cortex, and OFC (Jang and Kwon, 2014, “Neural Connectivity of the Amygdala in the Human Brain: A Diffusion Tensor Imaging Study”). To effectively explore this idea, we would want to effectively reconstruct these pathways in our sample using global and local tractography, which would be a considerable task. As such, we will endeavour to explore this question in a future study where we might take a specific focus on tractography-behaviour relationships across the cortex and subcortex.

For the present manuscript, we have included our thoughts on this point in the Discussion:

“Our findings open avenues for future studies on how this subcortical pathway might influence threat-related behaviour. While our findings demonstrated that greater pulvinar-amygdala fibre density related to better fearful face recognition, it remains to be seen how this might compare with structural connectivity of other cortical networks. In other words, would the fibre density of this subcortical connection explain fearful face recognition above and beyond, say, structural connections between the inferior temporal or orbitofrontal cortex and the amygdala (Pessoa and Adolphs, 2011) or between the thalamus and the superior temporal sulcus (Leppänen and Nelson, 2009)? Evidence from blindsight patients suggests that this subcortical connection ensures redundancy and compensation, such that it strengthens when cortical connections are destroyed (Tamietto et al., 2012). Taking this in conjunction with our findings, we might consider that the pulvinar-amygdala connection contributes to fear recognition in faces (and effective connectivity underlying face perception) in healthy participants but can increase or decrease its influence depending on the functioning of other networks. Such increases and decreases are already evident in certain clinical populations. For example, structural connectivity between the superior colliculus, pulvinar, and amygdala is weakened in individuals with autism compared to healthy controls (Hu et al., 2017), and BOLD signal to fearful faces is reduced in these areas (Kleinhans et al., 2011; Green et al., 2017), unless participants are explicitly instructed to fixate on the eyes (Hadjikhani et al., 2017). On the other hand, people who suffer from anxiety show hyperactive activity along the subcortical route compared to non-anxious individuals (Hakamata et al., 2016; Tadayonnejad et al., 2016; Nakataki et al., 2017). How and why this subcortical visual pathway to the amygdala is altered in these clinical populations remains a significant and relatively unexplored avenue of research.”

Minor Comments:1) The description of the behavioral task could be easier to discern by just stating, "To this end, we examined behavioral data from an out-of-scanner task, the Penn Emotion Recognition Task, that assessed a different component of face processing than the in-scanner task."

Thank you for the suggestion; this certainly makes the description clearer. We have added this to the manuscript.

“To this end, we examined behavioural data from an out-of-scanner task, the Penn Emotion Recognition task, that assessed a different component of face processing than the in-scanner task (analysed below). In the Penn Emotion Recognition task, participants were serially presented with 40 faces that were either happy, sad, angry, fearful, or neutral.”

2) Throughout the paper, there are a few instances of missing spaces between words and a parentheses and a missing period in subsection “Regions of Interest”.

Thank you, these have now been fixed.

3) The Results section employs suggestive language (“likely present”) while the Discussion is very strongly worded (“unequivocally supports”; “study settles”). Any chance for softening since all findings are at the limit of our current methods, so softening and adding some text about methodological considerations would strengthen the paper's ability to impact future work.

We agree with this suggestion. We have now softened the tone of the Discussion by removing several unnecessary adjectives:

“Our study, which used a large sample of participants from the HCP, supports the existence of a subcortical pulvinar connection to the amygdala in the healthy human adult brain that facilitates dynamic coupling between these regions and also enhances fear recognition.”

“We found that it was more likely for the network to include a subcortical visual route to the amygdala than a cortical route alone.”

“In conclusion, our study has made substantial progress towards settling the long-held debate over the existence and function of a subcortical route to the amygdala in the human brain.”

We have also added methodological considerations to the Discussion:

“Our decision to reconstruct the two halves of the subcortical route separately was motivated by our interest in the relative contribution of each connection to face-related processing (as described above) but was also a limitation imposed by anatomically-constrained tractography, where fibres terminate at boundaries between grey and white matter (Smith et al., 2012). Given that the pulvinar is made up of thalamic cell bodies (grey matter), the likelihood of reconstructing a continuous streamline of axon bundles traversing the pulvinar’s grey matter may have been restricted by these boundary constraints. Previous studies that have not imposed these constraints have successfully traced a continuous pathway from the superior colliculus to the amygdala via the pulvinar (Rafal et al., 2015; Tamietto et al., 2012), supporting animal research showing that inferior-lateral pulvinar neurons receiving superior colliculus afferents also have efferent connections to the lateral amygdala (Day-Brown et al., 2010). Our investigation into pulvinar and amygdala subregions support these findings, such that we found the superior colliculus to project predominantly onto the inferior (and anterior) pulvinar, which was the same subregion to receive the vast majority of fibres from the amygdala (see Figure 1). Furthermore, pulvinar fibres terminated predominantly within the basolateral amygdala, which is known to process visual information about threat and faces (Hortensius et al., 2016). Further studies could use both anatomically-constrained tractography and this subregion-specific approach with ultra-high-resolution imaging to better differentiate grey-white matter boundaries and more accurately determine if and where a continuous, subcortical route might traverse the pulvinar.”

“One limitation of the present study is the discrepancy between how local and global measures of fibre density related to other measures; namely, that local tractography covaried with fearful face recognition scores while global tractography covaried with effective connectivity. While the reconstructed fibres shared many similarities (e.g. the pattern of findings for each connection across hemispheres and subregions, as well as the overlap with major fasciculi; see Figure 1) even after accounting for head motion, it is possible that the local tractography’s relatively greater susceptibility to noise may have decreased its relationship to corresponding effective connectivity parameters. Indeed, global tractography has been shown to better reflect local connection architecture (Jbabdi and Johansen-Berg, 2011), such as the subcortical connections we have investigated. Such discrepancies between global and local tractography have been reported in other studies (Anastasopoulos et al., 2014) and so further research (particularly those that only recruit a single tractograpy method) will benefit from specific investigations into why these discrepancies might arise.”

Reviewer #3:

[…] I only have very minor comments ("minimal", I would say), to a paper already excellent.Minor Comments:I have only very minimal comments and suggestions that I would like the authors to take into consideration.First, it is interesting to note that across the different analyses there is a common trend about lateralization on this subcortical pathway in the right hemisphere. Originally, laterality in fMRI response in the amygdala and subcortical structures to masked facial expressions was reported in Morris et al., 1998 Nature and interpreted as supporting the role of the right hemisphere in non-conscious emotion perception. Anatomically, a similar trend in DTI data was reported in Tamietto et al., 2012, that, however, did not reach standard statistical threshold. I would invite the authors to comment a bit more on this laterality effectin relation to previous works and neuropsychological evidence.

We agree that this is an interesting aspect of our results and is worth commenting on. We have now included this point in the Discussion:

“We observed hemispheric lateralisation of the pulvinar-amygdala connection, such that both the local and global tractography showed greater fibre density along the right than the left, and there were stronger tractography-behaviour and tractography-connectivity relationships for the right than the left. Early studies on the subcortical route observed specifically right-sided BOLD responses during non-conscious fearful face viewing (Morris et al., 1999; Morris et al., 1998), and a previous tractography study has also found that only the fractional anisotropy of the right subcortical route was significantly related to threat-biased (Koller et al., 2018). There is mounting evidence for right-sided specialisation for ordered (Wyczesany et al., 2018) and disordered (McDonald, 2017) emotion processing, particularly for non-conscious signals transmitted along the subcortical route (Gainotti, 2012). Thus, our results lend support to this theory by demonstrating evidence for the right pulvinar-amygdala connection’s stronger fibre density and its relationship to emotional face viewing and fearful face recognition. Our understanding of this lateralisation may be deepened by future exploration of left- vs. right-sided structural connectivity and function along the subcortical route during conscious vs. non-conscious emotion processing in healthy participants.”

Second, there is recent and converting evidence in patients with V1 damage and "affective blindsight" that this subcortical pathway seems tuned to process low-spatial frequency information (e.g., for a recent paper Burra et al., 2018). Evidence in healthy participants seems more controversial, with some findings supporting a similar specialization for low spatial frequencies (e.g., Méndez-Bertolo et al., 2016, Carretié et al., 2017 JoCN), and other data, eminently from the same authors purporting a more "generalized" role (McFadyen et al., 2017). Possibly, some further speculation and more explicit reference to this topic would be of interest in the light of the new data.

Thank you for drawing our attention to this. Our findings do indeed contribute to this issue by demonstrating that:

a) The tractographic reconstruction of the subcortical route presented the inferior-lateral-anterior pulvinar as a likely connection point between the superior colliculus and amygdala, suggesting continuity of the full pathway (note that this is new data added in light of Reviewer 1’s comments – see the updated Figure 1)

b) Cortical-pulvinar connections (i.e. inferior occipital gyrus to pulvinar) were not present in the winning dynamic causal model, suggesting that the pulvinar-amygdala connection was not driven by occipital input.

These two points favour the hypothesis that the pulvinar-amygdala transmission was driven by input from the superior colliculus and thus may consist of predominantly low spatial frequencies. Evidence against this hypothesis comes from the fact that the winning dynamic causal model included input to both the superior colliculus and input directly to the pulvinar, the latter of which could reflect direct retinal input or input from (potentially parvocellular) areas not explicitly defined in the model (e.g. extrastriate cortex, LGN, temporal cortex excluding the fusiform gyrus, etc.). Hence, our study elucidates many aspects of the pulvinar-amygdala connection but was not able to fully address the precise sources of input (and their theoretical spatial frequency tuning) to this connection beyond the superior colliculus. We have incorporated these points into the Discussion:

“While our results suggest that the inferior pulvinar may serve as a disynaptic connection point between the superior colliculus and amygdala, the continuity of information flow along the subcortical route is still a disputed feature due to the strong cortical influences on the pulvinar (Bridge et al., 2016; Pessoa and Adolphs, 2011). This dispute has also arisen from prior work investigating the spatial frequency content of information conveyed along the subcortical route. Research on blindsight patients has found evidence only for low spatial frequencies which suggests that such information originated from magnocellular cells in the superior colliculus (Burra et al., 2018; Méndez-Bertolo et al., 2017). On the other hand, work in healthy participants has found no such spatial frequency preference, which suggests that rapid pulvinar-amygdala transmission might include input from other parvocellular pathways (McFadyen et al., 2017). We did not exhaustively explore the extent to which the cortex contributes information to the pulvinar-amygdala connection. The winning effective connectivity model, however, did not include cortical connections between the pulvinar and the inferior occipital gyrus. Hence, it is unlikely that the primary visual cortex contributed (either via direct anatomical connections or functional coupling along the ventral visual stream (Pessoa and Adolphs, 2010) to the information transmitted along the subcortical route. The winning model did, however, include input to the superior colliculus as well as directly to the pulvinar, which could reflect direct retinal input or input from areas not explicitly included in the model, such as the (e.g. parietal cortex, temporal cortex, or the LGN; Bridge et al., 2016), that may transmit both low and high spatial frequency information. Furthermore, it remains to be shown how interactions between the pulvinar and other cortical areas, such as the inferotemporal cortex (Zhou et al., 2016), may directly influence activity along the pulvinar-amygdala connection.”

Third, one strong hypothesis verified in non-human models is that the subcortical route to the AMG is composed by a functionally integrated disynaptic pathway whereby the superficial layers of the SC project to infero-later PULV and the same neurons in the PULV then send efferents to the lateral AMG (a summary of evidence about different subcortical or non-canonical pathway to the amygdala in humans and non human animals can be found in Diano et al., 2017 Front Psychol). In an attempt to provide indirect evidence in humans, some previous tractography studies, after tracing fiber connection between SC and PULV, and between PULV and AMG, also tried to verify whether a subsample of these reconstructed fibers could be considered as belonging to the same streamlines. If I am not missing something, the present analyses do not investigate this aspect. It would be nice to have this info if possible. I acknowledge the paper is already methodologically rich, so I am not explicitly asking additional analysis, even though this should not be particularly long to perform. However, mentioning to this aspect for future development would be appropriate.

Thank you for raising this point. We actually did initially reconstruct fibres for the full SC-PUL-AMG pathway (using either SC or AMG as seed regions and PUL as an intermediate region) but found their numbers to be considerably lower than those of the independent SC-PUL and PUL-AMG pathways (see Author response image 1). We then realised that the nature of our tractography algorithm may be restricting the number of successful streamlines for the full subcortical route – that is, the tractography was anatomically constrained, meaning that streamlines terminated if they reached a grey-matter boundary. Given that the pulvinar constitutes a bundle of grey matter imaged at 1mm voxel resolution, it may have been difficult for the algorithm to successfully perpetuate a streamline of bundled axons (white matter) through the pulvinar ROI *without* encountering a grey matter voxel (see example of one participant’s tissue segmentation of the pulvinar in Author response image 1). Interestingly, though, the fibre density was still markedly low even when we removed these anatomical constraints (see Author response image 1), indicating that we could not frequently trace continuous bundles of axons between SC and AMG that penetrate the pulvinar using this sample of diffusion images. Hence, our decision to reconstruct the two halves of the subcortical route separately.

Additionally, we believe that the findings presented for the SC-PUL and PUL-AMG connections (which would include any fibres that traverse the entire SC-PUL-AMG pathway) yield even more detailed information by demonstrating the relative contribution of the two halves of this pathway. We have now included these points in the Discussion:

“Our decision to reconstruct the two halves of the subcortical route separately was motivated by our interest in the relative contribution of each connection to face-related processing (as described above) but was also a limitation imposed by anatomically-constrained tractography, where fibres terminate at boundaries between grey and white matter (Smith et al., 2012). Given that the pulvinar is made up of thalamic cell bodies (grey matter), the likelihood of reconstructing a continuous streamline of axon bundles traversing the pulvinar’s grey matter may have been restricted by these boundary constraints. Previous studies that have not imposed these constraints have successfully traced a continuous pathway from the superior colliculus to the amygdala via the pulvinar (Rafal et al., 2015; Tamietto et al., 2012), supporting animal research showing that inferior-lateral pulvinar neurons receiving superior colliculus afferents also have efferent connections to the lateral amygdala (Day-Brown et al., 2010). Our investigation into pulvinar and amygdala subregions support these findings, such that we found the superior colliculus to project predominantly onto the inferior (and anterior) pulvinar, which was the same subregion to receive the vast majority of fibres from the amygdala (see Figure 1). Furthermore, pulvinar fibres terminated predominantly within the basolateral amygdala, which is known to process visual information about threat and faces (Hortensius et al., 2016). Further studies could use both anatomically-constrained tractography and this subregion-specific approach with ultra-high-resolution imaging to better differentiate grey-white matter boundaries and more accurately determine if and where a continuous, subcortical route might traverse the pulvinar.”

There is a factual error in the Introduction section where authors say about Tamietto et al., 2012 paper that "The white matter structure of the subcortical route was estimated for the patient and for six healthy, age-matched controls". In fact, the healthy controls analyzed were 10, not 6.

This has now been corrected.